# Benefits of a robotic chamber system for determining evapotranspiration in an erosion affected, heterogeneous cropland

Adrian Dahlmann[1], Mathias Hoffmann[1], Gernot Verch[2], Marten Schmidt[1], Michael Sommer[3,4], Jürgen Augustin[1], Maren Dubbert[1]

[1]Isotope Biogeochemistry and Gas Fluxes, Leibniz Centre for Agricultural Landscape Research, Müncheberg, 15374, Germany
[2] Experimental Infrastructure Platform (EIP), Leibniz Centre for Agricultural and Landscape Research, Prenzlau, 17291, Germany
[3]Landscape Pedology, Leibniz Centre for Agricultural Landscape Research, Müncheberg, 15374, Germany
[4] Institute of Geography and Environmental Science, University of Potsdam, 14476, Potsdam, Germany

*Correspondence to*: Adrian Dahlmann (adrian.dahlmann@zalf.de)

**Abstract.** In light of the ongoing global climate crisis and related increases in extreme hydrological events, it is crucial to assess ecosystem resilience and - in agricultural systems - to ensure sustainable management and food security. For that purpose, comprehensive understanding of ecosystem water cycle budgets and spatio-temporal dynamics are indispensable. Evapotranspiration (ET) plays a pivotal role returning up to 90 % of incoming precipitation back to the atmosphere. Here, we studied impacts of soil types and management on an agroecosystem's seasonal cumulative ET ($ET_{sum}$) and agronomic water-use efficiencies ($WUE_{agro}$, dry matter per unit of water used by the crop). To do so, a plot experiment with winter rye (September 17, 2020 to June 30, 2021) was conducted at an eroded cropland which is located in the hilly and dry ground moraine landscape of the Uckermark region in NE Germany. Along the experimental plot (110 m x 16 m), two closed chambers were mounted on a robotic gantry crane system (FluxCrane as part of the AgroFlux platform) and used to determine ET. Three soil types representing the full soil erosion gradient related to the hummocky ground moraine landscape (extremely eroded: Calcaric Regosol, strongly eroded: Nudiargic Luvisol, non-eroded: Calcic Luvisol) and additional topsoil dilution (topsoil removal and subsoil admixture) were investigated (randomized block design, three replicates per treatment). Five different modeling approaches were used and compared in light of their potential for reliable $ET_{sum}$ over the entire crop cultivation period as well as to reproduce short-term (e.g., diurnal) water-flux dynamics. While machine learning approaches such as support vector machines (SVM) and artificial neural networks (with Bayesian regularization; ANN_BR) generally performed well during calibration, SVM also provided a satisfactory prediction of measured ET during validation (k-fold cross validation, k = 5). We found significant differences in dry biomass (DM) and small trends in $ET_{sum}$ between soil types, resulting in different $WUE_{agro}$. The extremely eroded Calcaric Regosol showed an up to 46 % lower $ET_{sum}$ and up to 54 % lower $WUE_{agro}$ compared to the non-eroded Calcic Luvisol.  The key period contributing to 70 % of $ET_{sum}$ spanned from the beginning of stem elongation in April to harvest in June. However, differences in the $ET_{sum}$ between soil types and topsoil dilution resulted predominantly from small differences between the treatments throughout the cultivation, rather than only during this short period of time.

# 1 Introduction

Only 12 % of the world's land area is suitable for food and fiber production due to its highly productive soils (Blum, 2013). Much of this land is already in use to ensure food security, mandated by a still growing human population paired with the ongoing climate crisis (Searchinger et al., 2018). Worldwide, land area is largely affected by soil degradation (Jie et al., 2002) and agriculture is closely related, since at least six degradation processes (e.g., erosion or compaction) are associated with it (Louwagie et al., 2011). In hummocky landscapes, erosion and associated topsoil dilution caused by, e.g., wind, water or tillage

affects the crop yields (Bakker et al., 2007; Biggelaar et al., 2003). In addition, weaker rootability on eroded soils suggests a higher susceptibility towards droughts (Schneider and Don, 2019). However, methodologically studying the influence of small-scale soil heterogeneity (e.g., soil erosion) and land use (e.g., soil management) on the dynamics of the water balance (especially evapotranspiration (ET)) separately has been challenging. The effect of both factors can be significantly different with complex interactions, e.g., soil erosion can lead to differences in soil water storage capacity and management affects soil

organic matter and water retention (Bakker et al., 2007; Biggelaar et al., 2003). Thus, a separate response analysis is an indispensable prerequisite for the development of site-specific land use procedures adapted to the changing climate conditions. Moreover, the climate crisis is affecting the amount and spatio-temporal distribution of precipitation worldwide, leading to more frequent and stronger precipitation events in high-precipitation regions (e.g., increase of 10 – 40 % in northern Europe; (DWD, 2019) and fewer and weaker events in low-precipitation regions (e.g., up to 20 % decrease in the Mediterranean region

and southeastern Europe; Trenberth, 2011). In Germany, annual precipitation is more than 800 mm in most regions of west and south Germany but only 400 - 500 mm $y^{-1}$ in the northeast (e.g., areas in Brandenburg and Mecklenburg-Western Pomerania; Schappert, 2018). Here, dry hydrological conditions and erosion shaped landscapes meet. As crop yields and related crop productivity depend on various factors such as soil properties or water availability, such agriculturally used precipitation limited regions could face increasing problems.

ET describes the total amount of water that evaporates from a given area and is thus defined as the sum of soil evaporation (E), transpiration (T) and interception evaporation (Fohrer et al., 2016; Rothfuss et al., 2021). Generally, ET is one of the most important components of the hydrological cycle in terrestrial ecosystems, accounting for up to 100 % of ingoing precipitation (Hanson 1991). Due to the expected increasing dependency of a system's productivity on sufficient water supply with an accelerating climate crisis, quantifying the ET plays an important role to achieve a process-based understanding of the

mitigation potential of different crops to drought in the future and to, e.g., establish a more efficient supplemental irrigation. Moreover, there is a tight link of carbon and water cycling in precipitation limited systems because water loss by ET and the water use efficiency of a system can largely define its productivity (Tallec et al., 2013).

A particular challenge in current ET research is combining high-frequency with multi-treatment approaches. At the field scale for example, eddy covariance systems provide high-frequency estimates of ET of a homogeneous system (e.g., Ding et al.,

2021), while manual chamber approaches are able to precisely capture multi-treatment effects (<1m²) on ET at the plot scale (e.g., Hamel et al., 2015). However, the data frequency obtained by manual chamber observations is often too small to achieve

reliable flux budget calculations in combination with data-driven modeling. In this regard, modern automated chamber systems allow a combination of multi-treatment observation with a higher frequency of measurements compared to manual chambers. They provide a unique opportunity to test data-driven modeling strategies, using empirical, statistical, or machine learning approaches, with the aim of reproducing the diurnal variability in ET and the seasonal cumulative ET ($ET_{sum}$). In particular, modeling approaches based on machine learning (e.g., artificial intelligence and neural network approaches) have previously been limited to eddy covariance measurements. Coupling such advanced modeling strategies with modern automatic chamber systems might be an ideal fusion of measurement frequency and the ability to capture treatment effects like small-scale soil differences (Falge et al., 2001a; KIŞI and ÇIMEN, 2009). AgroFlux – a newly-developed sensor platform centered around closed chambers mounted on an robotic gantry crane (FluxCrane) – was initially built to capture the effect of soil type and management on GHG emissions and in particular $CO_2$ fluxes with high spatial and temporal resolution (Vaidya et al., 2021). The adaption of the system to measure ET provided us with the opportunity to analyze stand scale ET fluxes including the development of a data analysis tool for measured ET fluxes, and to test different modeling strategies. We tested five different modeling strategies including basic statistic and advanced approaches including machine learning approaches. During the cultivation period of winter rye from mid-September 2020 to the end of June 2021, ET and relevant environmental and plant growth parameters were measured to identify the corresponding drivers of crop ET and productivity. The FluxCrane system covers a field where three different soil types are present, which reflect the erosion gradient typical for the hillside of the hummocky ground moraine landscape of northeast Germany. This made it possible to evaluate the impact of soil type as well as soil management on $ET_{sum}$, seasonal development and agronomic water use efficiency ($WUE_{agro}$; dry matter per unit of water used by the crop).

In the following, we will examine i) soil type and topsoil dilution effects on crop yield, $ET_{sum}$ and $WUE_{agro}$, ii) the spatio-temporal variability of ET fluxes over the growing season, and iii) the suitability of various modeling strategies. The paper's aim is to establish an approach that would provide reliable predictions of ET fluxes both in terms of $ET_{sum}$ as well as diurnal trends of ET fluxes. We hypothesize that: i) eroded soils and topsoil dilution lead to decreased ET controlled by weaker plant growth, ii) $WUE_{agro}$ declines from least to most eroded soil type and with topsoil dilution; iii) the automated, continuous FluxCrane measurements result in unique insights into smallscale dynamics such as night time ET fluxes and ET fluxes during the non-growing season. Here, we hypothesize, that iv) the uncommonly large data set (compared to manual chamber systems) allows for a robust use of modeling strategies based on machine learning. We envisage that this will greatly improve $ET_{sum}$ and subsequently $WUE_{agro}$ based on automated closed chamber systems.

## 2 Material and Methods

### 2.1 Study site and experimental design

The AgroFlux experimental platform is located in Brandenburg, a federal state in northeast Germany, near Dedelow within the Uckermark region (53° 23´ N, 13° 47´ E; ~50-60 m a.s.l). Brandenburg, which includes some of the driest regions in Germany, uses 48.6% or about 1.44 million hectares of its area for agriculture (Amt für Statistik Berlin-Brandenburg, 2022). It is located in the continental climate zone and has a water deficit of about 150 mm during the growing season (Wessolek and Asseng, 2006). The long-term (1991 to 2020; ZALF) mean annual air temperature in this region is 8.8°C with a mean annual precipitation and potential evapotranspiration of 467 mm and 637 mm, respectively (ZALF research station, Dedelow). The focus of agriculture in Brandenburg is on grain production, which faces a variety of challenges due to increasingly dry conditions during the main growing season (Amt für Statistik Berlin-Brandenburg, 2022). The Uckermark region is the most productive region for agriculture within Brandenburg. It is shaped by glaciation with a hilly to flat-wavy ground moraine landscape whose soils are strongly influenced by soil erosion (Nudiargic Luvisol, Calcaric Regosols, Colluvic Regosols) as well as redoximorphic soils (Stagno-, Gleysols) (MLUK, 2020). The strong soil heterogeneity and ongoing soil erosion, mainly by tillage, has a great influence on the productivity of the cultivated areas (Sommer et al., 2016). Today, only 20% of the land is not affected by past and present soil erosion due to tillage and water (Sommer et al., 2008; Wilken et al., 2020), resulting in a very high spatial variability of soils (Wehrhan and Sommer, 2021) and associated growing conditions for crops (Wehrhan et al., 2016). In combination with the ongoing climate crisis, it is proving difficult to develop land-use methods that allow reliable and sustainable arable farming under these challenging conditions.

The study was carried out on the 100 x 16 m FluxCrane experimental field, an integral part of the AgroFlux sensor platform located at the interdisciplinary research area CarboZALF-D (Fig. 1a). There is an elevation difference of one meter and all relevant local erosion stages are covered (Schad, 2016): non-eroded Calcic Luvisol (LV-cc), strongly eroded Nudiargic Luvisol (LV-ng) and extremely eroded Calcaric Regosol (RG-ca; see Fig. 1b/c; Sommer et al., 2008; Wehrhan et al., 2016; Vaidya et al., 2021). Here we used 18 plots in total, six per soil type (Fig. 1d). For the six plots per soil type, a randomized, full-factorial design, each repeated threefold, was adopted for topsoil dilution vs. non-topsoil dilution (first 8 to 9 cm). During the study period from September 2020 to June 2021 (286 days), winter rye of the hybrid variety SU Piano was grown with a density of 200 plants per m² on an area of 0.176 ha. The novel gantry crane automatic chamber system (Fig. 1e) was installed on this study site in 2019 (Vaidya et al., 2021). The attached gas exchange chambers were lowered on each plot on round structural steel frames with a diameter of 1.59 m and a basal area of 1.99 m$^2$.

### 2.2 Cultivation and topsoil dilution

The AgroFLUX sensor platform site is located on a conventionally farmed agricultural area that is intended to represent a variety of soils in the region. Hence, topsoil dilution, tillage, cultivation and fertilizer application measures were implemented before and during the experiment. The manipulative field experiment was originally established to study the feedbacks of a

dynamic disequilibrium in the carbon cycle of arable lands. Deep tillage or soil erosion lead to an admixture of subsoil material into the plough layer (Doetterl et al., 2016) which alters topsoil properties (SOC, clay content etc.). The resulting changes in the main rooting zone might reduce crop growth (Öttl et al., 2021). We mimic these common landscape processes in our topsoil dilution experiment under controlled conditions (Vaidya et al., 2021). After topsoil removal (1.2 t per plot; first 8-9 cm; 3 of the 6 plots per soil; July 14-15, 2020) we added the equivalent mass (1.2 t) of the respective subsoil horizons (E, Bt, Ck) taken from a large soil pit nearby. Thus, E horizon was applied to the prepared plots of the non-eroded Calcic Luvisol (LV-cc), Bt horizon on the strongly eroded Nudiargic Luvisol (LV-ng) and Ck horizon to the extremely eroded Calcaric Regosol (RG-ca). Subsequently, we mixed the added subsoil material with the remaining local Ap horizon. Finally, the chamber frames were reinstalled. In the following, the resulting treatments of the same soil types are labelled as non-diluted (n-d) and diluted (d). The actual tillage prior to sowing took place just before seeding on September 17, 2020. For this, the frames were removed, and the soil was loosened to a depth of 25 cm in west-east-direction. Sowing was done with a power harrow-drill combination. Fertilization was applied to all plots per soil type before and during the growing season using digestate from Pflanzenbauhof GbR (Uckermark, Germany), Triple Super Phosphate (TSP) and grain potash (Table B1). Due to initial changes in the topsoil structure (after the addition of subsoil material), germination differed between manipulated and non-manipulated plots. In order to achieve similar plant densities in all plots, replanting had to be done in all n-d plots within the frames (LV-cc: 13 plants per plot; LV-ng: 40 plants per plot; RG-ca: 82 plants per plot). For general plant protection and soil treatment, herbicides were applied to the field prior to the growing season (e.g., glyphosate; September 3, 2020).

## 2.3 Gantry crane system description and gas exchange measurements

The ET flux measurements were carried out by a novel automated chamber system (FluxCrane) using a 5-meter-high gantry crane traveling on two 110 m tracks which has been described in detail (Vaidya et al., 2021). Briefly, the system designed by Pfannenstiel ProProject GmbH (Bad Tölz, Germany) is capable of moving in three dimensions: the x-axis for movement along the track, the y-axis for movement perpendicular to the track, and the z-axis for vertical chamber movement. The FluxCrane carries two transparent chambers made of polymethyl methacrylate (PMMA; A: 1.986 m$^2$; V: 3.756 m$^3$). Since the two chambers do not move independently from each other along the track, frames were arranged in rows, from which each half was measured by one chamber. To ensure airtight sealing during chamber deployment, steel frames with a diameter of 1.59 m and a depth of 5 cm were installed into the soil and equipped with an approximately 10 cm wide foam ring to further increase the chambers bearing surface while deployed. ET fluxes were determined by measuring the development of chamber headspace H$_2$O concentrations (4 sec frequency) over 7 minutes in a flow-through non-steady-state (FT-NSS) mode (Livingston and Hutchinson, 1995), using two infrared gas analyzers (one per chamber; LI-COR 850, Licor Biosciences, UK). The chambers have an average light transmittance of about 76 % (74% for chamber 1 and 78% for chamber 2), but the experiment was designed to minimize a reduction in ET due to reduced light availability (Fastest possible ET calculation after chamber closure; short closing times and ventilation). In addition, Pape et al. (2009) concluded, that the PAR reduction had only a minor effect (< 5%) on photosynthesis for this type of chamber, which should be comparable or smaller for transpiration. Temperature

differences during chamber closure were minimized by the short measurement time and ventilation (<1.5°C) with two small axial flow fans (5.61 m$^3$ min$^{-1}$) used to homogenize the chamber headspace air. To compensate for the difference in tubing length between the chambers and the analyzer (chamber 1: 15 m vs. chamber 2: 22 m), a flow rate of 2.3 l min$^{-1}$ and 3.6 l min$^{-1}$ was set to obtain a similar sensor death time of 13 seconds. A CR6 data logger and a CDM-A116 analog multiplexer (Campbell Scientific Inc., USA) were used for data recording and storage. The plots were measured hourly up to 24 times a day in order to be able to detect daily variations. Due to the randomized measurement design, each plot was measured approximately twice per week, which would theoretically result in approximately 2200 measurements per plot throughout the entire season. However, the system was designed to shut down under high winds and cold temperatures, resulting in a true average of only 724 measurements per plot per season. Diurnal ET day- and nighttime fluxes considered in this study were calculated for the cultivation period from September 17, 2020 (sowing of winter rye), until harvest of winter rye on June 30, 2021.

## 2.4 Input parameters for modeling

### 2.4.1 Environmental parameters

Relative humidity (RH) [%] (WXT520, Vaisala, FI) was measured during the ET flux measurements outside the chambers while temperature (T) [°C] (109, Campbell Scientific Ltd., USA) and incoming photosynthetically active radiation (PAR) [µmol m$^{-2}$ s$^{-1}$] (SKP 215, Skye Instruments Ltd., UK) were measured both outside as well as inside the chambers. Precipitation (Pr) [mm] (Tipping Bucket Rain Gauge 52203, R. M. Young Company, USA) and relative soil moisture (SM; 13 to 18 cm depth) [%] (ML2x, Delta-T Devices Ltd., UK) were measured at an adjacent field (< 25m; Fig. 1b).

### 2.4.2 Plant-specific parameters

Spectral plant indices, such as the ratio vegetation index (RVI; also simple ratio SR) were manually recorded weekly for all 18 plots using a near-infrared (NIR)/visible light (VIS) double, 2 channel sensor device (SKR 1850, Skye Instruments Ltd., UK) mounted on a 1.8 m handheld pole (Görres et al., 2014; Kandel et al., 2013), connected to a CR1000 data logger (Campbell Scientific Ltd., USA). The double, 2 channel sensor device consisted of an upward- and downward–facing sensor, measuring the incoming (VISi) and reflected (VISr) VIS at a wavelength of 656 ± 10 nm and incoming (NIRi) and reflected (NIRr) NIR at 780 ±10 nm. The upward sensor was fitted with a cosine-correction diffusor for measurements of the incident radiation, while the downward sensor, installed 1.8m above the ground, had a 25° cone field of view, thus covering an area of 0.5 m$^2$ during measurements (Görres et al. 2014). Each plot was measured once a week for 30 seconds, resulting in one mean value including 30 measurement points. The RVI was used as an indicator for standing crop biomass and is close to zero for a fallow surface and increases as plant cover increases. The RVI was calculated following Equ. 1:

$$RVI = \frac{\frac{NIRr}{NIRi}}{\frac{VISr}{VISi}} \qquad (1)$$

Since only weekly plot-wise RVI data was available, daily RVI data was obtained by fitting a sigmoidal function for initial plant growth in the fall up to stagnation due to plant inactivity in the winter and a polynomial function for shoot elongation and later senescence during spring growth and summer maturation, respectively (Fig. A1). During the period from November 24, 2020 to March 22, 2021, which we refer to as the non-growing season, no plant growth was assumed due to average daily temperatures below 5°C (<3 consecutive days).

## 2.5 ET flux calculation and modeling

### 2.5.1 ET flux calculation

The workflow included various steps to pre-process data obtained by the FluxCrane, calculate ET fluxes and finally applying and validating the different modeling procedures (Fig. A2). ET flux calculation was performed based on the ideal gas equation (Eq. 2) modified by (Hamel et al., 2015) using an adapted R-script, based on those presented by (Hoffmann et al., 2015).

$$ET_{flux} = \frac{c_{H2O} \times P \times M_{H2O}}{R \times T} \times \left(t \times \frac{V}{A}\right)\left[\frac{mm}{d}\right] \tag{2}$$

$$ET_{flux}[mmol\ m^{-2}s^{-1}] = \frac{ET_{flux}\left[\frac{mm}{d}\right]}{(t \times 1000)} * \left(\frac{1}{M_{H2O}}\right) \tag{3}$$

With $ET_{flux}$ [mm d$^{-1}$] being the evapotranspiration rate, $c_{H2O}$ the moles of water per total moles present, P the gas pressure [Pa], $M_{H2O}$ the molar mass of water [18 g mol$^{-1}$], R the gas constant [8.314 m³ Pa K$^{-1}$ mol$^{-1}$], T the temperature [K] inside the chamber, t the time factor [86.4], V the chamber volume [m³] and A the basal area [m²]. The ET flux in mmol m$^{-2}$ s$^{-1}$ (Equ. 3) was also calculated to ensure comparability with other studies. The first 15% of each measurement was discarded prior to flux calculation to prevent a disturbance due to initial homogenization of the chamber headspace air. The temporal change was determined by linear regressions on several subsets of the data generated based on a variable moving window with a starting window size of 1:20 minutes (20 consecutive data points) and a maximum length of 2 minutes (30 consecutive data points). This procedure resulted in several ET fluxes for each measurement, from which the best flux was subsequently selected using a set of soft and hard criteria. Soft criteria included: (i) checking whether the conditions for the application of a linear regression were fulfilled (normality, variance homogeneity, linearity); (ii) no outliers were present (±6x interquartile range); (iii) temperature variation during the measurement was < 1.5 °C. Calculated fluxes per measurement that did not meet the quality criteria were discarded. Afterwards applied hard criteria reduced potentially remaining multiple fluxes per measurement further to the ideal ET flux. Since the air in the chamber headspace reached higher water saturation with increasing time, hard criteria were based on the selection of the flux which showed the shortest temporal distance to the start of measurement and had the maximum length.

During the measurements, various events could lead to erroneous ET fluxes such as e.g., fog (saturation of the chamber interior), sensor failures, or chamber leakage due to failure in chamber deployment. Erroneous fluxes were hence discarded.

In addition, potential differences of the measurements between the sensors of both chambers were evaluated by the measurements of ambient air during periods of no chamber deployment.

A complete data set of hourly data points for the 286 days of the cultivation period would consist of 6,888 measurements per treatment. After the flux calculation and filtering using the soft and hard criteria, a total of 13,011 ET flux measurements were performed, resulting in approximately 2,169 measurements per treatment. On individual plots, an average of 723 measurements were available (Table B2; ranging from 624 to 1,210; accounting for 10.5 % on average).

### 2.5.2 Modeling ET fluxes

To model ET fluxes, five different approaches were used and compared with each other. Modeling procedures included: 1.) a simple statistical approach: Mean diurnal variation (MDV); two empirical approaches: namely 2.) non-linear regression (NLR) and 3.) Look-Up-Tables (LUT) as well as two machine learning approaches: with 4.) Support Vector machine (SVM) and 5.) artificial neural network with Bayesian regularization (ANN_BR).

MDV (Falge et al., 2001b; Moffat et al., 2007) is used to calculate missing hourly values through interpolation of values measured at the same hour during adjacent days. Thus, for the season with 286 days, the missing values were calculated for every hour, generating 24 values per day.

NLR is based on parameterized non-linear equations using the mean least square method to express the relation between the total seasonal data of ET and T, RH, SM, PAR and RVI. Half-hourly values were predicted using the predictor variables and obtained function parameters.

Modeling missing ET fluxes using the LUT approach is based on the assumption of similar ET fluxes during similar environmental conditions, whereby similarity is defined through a number of thresholds for the different environmental variables. Thus, missing ET-fluxes can be predicted based on the environmental conditions as well as the RVI associated with the missing data. To do so, measured ET-fluxes per subplot were split into different classes ($c_{sturges}$) based on T, RH, SM, PAR and RVI, with their class limits determined by Sturges rule (Equ. 4, Sturges, 1926). Within this study, on average, 12 classes of equal size were formed covering the range of all parameters.

$$c_{sturges} = \frac{1 + 3.32 * log\,(n)}{log\,(10)} \tag{4}$$

ET fluxes were subsequently assigned with the average ET flux of the class corresponding to obtained environmental parameters. In case no class could be attributed to measured environmental conditions, the average ET flux was used.

SVM is a black-box model, where a computer algorithm learns by teaching data to assign values to objects or classes (Noble, 2006). As mentioned by Kim et al. (2020), the SVM uses a slack variable to circumvent unseparated parameters due to noise or extreme values in the data, as well as the radial basis kernel function based on previous SVM studies for upscaling fluxes (Ichii et al., 2017; Xu et al., 2018).

In comparison, ANN_BR is a combination of a purely empirical nonlinear regression model with a stochastic Bayesian algorithm for regularizing the network training (Schmidt et al., 2018). An artificial neural network (ANN) consists of nodes connected by weights representing the regression parameters (Bishop, 1995; Hagan et al., 1996; Moffat et al., 2007; Kubat, 1999; Rojas, 1996). The network is trained by providing it with sets of input data such as the environmental and plant-specific parameters mentioned earlier and the associated output data in the form of e.g., ET flux values. Similar to Moffat et al. (2007), all techniques evaluated use the classical back-propagation algorithm, where the training of the ANN is performed by propagating the input data through the nodes via the weighted connections and then back-propagating the error and adjusting the weights so that the network output optimally approximates the ET-fluxes. Subsequent to this training, the underlying dependencies of the ET fluxes on the environmental and plant-specific input variables are mapped to the weights and the ANN is used to predict half-hourly ET fluxes, where the performance of the ANN depends on several criteria.

## 2.6 Seasonal cumulative ET, water use efficiency and crop ET

$ET_{sum}$ were determined using half-hourly or hourly ET values predicted by all five modeling approaches. Daily averages [mm $d^{-1}$] and $ET_{sum}$ (mm cultivation period$^{-1}$) were formed in order to view the development over the entire cultivation period. The amount of plant biomass in dry mass (DM) [kg] was recorded during harvest for each treatment, which, in combination with $ET_{sum}$ yields the agricultural water use efficiency $WUE_{agro}$ (Hatfield and Dold, 2019; Equ. 5). This is the $WUE_{ABG}$ variant of $WUE_{agro}$, as the dry mass is total aboveground biomass (Katerji et al., 2008).

$$WUE_{agro} = \frac{DM}{ET_{sum}} \tag{5}$$

To obtain a comparable value for the $ET_{sum}$ calculated by the FluxCrane, crop evapotranspiration ($ET_c$) was calculated (Allen 1998). $ET_c$ (Equ.6) was calculated from the crop factor $K_c$ ($K_{c\ ini}$ = 0.3; $K_{c\ mid}$ = 1.15; $K_{c\ end}$ = 0.4) and the potential evapotranspiration $ET_0$ using monthly averages (DWD, 2022).

$$ET_c = K_c \times ET_0 \tag{6}$$

## 2.7 Statistical analysis

All calculations were performed using the statistical software R (R Core Team, 2021) version 4.0.4. Therefore, several packages (Table B3) were used to calculate the ET fluxes and to perform subsequent modeling as well as for the visualization of results.

To calibrate the modeling approaches, a comparison was conducted between all measured values and their corresponding predicted values for each treatment. For validation, the k-fold cross method (k = 5) was implemented using the resulting ET data to evaluate the predictive performance of the approaches and ensure robust statistical analysis. To accomplish this, each

data set was divided into five subsets, each comprising 20% of the total data. The modeling process was then repeated five times, utilizing 80% of the data to calculate the missing 20% and generate a complete dataset without relying on the original data. Subsequently, this dataset was compared against the measured data to evaluate the modeling approaches. Finally, performance metrics including the coefficient of determination (R2), mean absolute error (MAE), normalized root mean square error (NRMSE) and Nash-Sutcliffe efficiency (NSE) were calculated for both calibration and validation. These metrics were

used to define performance classes (Table 1) for evaluating the accuracy of the approaches in the given setup (Moriasi et al., 2015). To determine parameter impact on ET, linear and non-linear models were used. Lastly, differences of $ET_{sum}$, DM and $WUE_{agro}$ between treatments were tested with the Kruskal-Wallis-test and Dunn-Bonferroni post-hoc test.

## 3. Results

### 3.1 Environmental parameters

The study year was significantly warmer (mean temperature 9.6 °C) and wetter (508 mm annual precipitation) between July 1, 2020 and June 30, 2021, compared to mean annual air temperature (8.8 °C) and precipitation (467 mm). In particular, temperatures (Fig. 2a) were above average in the fall and winter period in 2020 as well as June, 2021. On the other hand, April and May, which are crucial for crop growth, were colder and also drier. Daily mean RH (Fig. 2b) ranged between 50 % and 92.4 % with increasing diurnal variation in warm periods. PAR (Fig. 2c) largely reflected the seasonal variation of the day

length with a maximum of 1860 µmol m$^{-2}$ s$^{-1}$ (half-hourly measurements), and reduced values during longer storm events and high cloud cover (e.g., through changes in photosynthesis). The SM at 13 to 18 cm depth largely reflects the intensity of precipitation events (Fig. 2d), ranging from 12 % to 29 %. One exception is a prominent increase in mid-February that can be attributed to low temperatures and subsequent snowmelt. The largest precipitation events (> 10 mm d$^{-1}$) occurred on September 26, 2020 with 12 mm, on December 24, 2020 with 15 mm and on February 3, 2021 with 16 mm. A sharp declining trend in

SM and no response to precipitation events is evident from April (about 25 %) to harvest in June (about 12 %). However, this can be explained by a high water consumption of the fully developed crop stand and canopy interception. Shallower SM sensors at 3 to 8 cm (not shown) indeed responded to these precipitation events albeit weakly, indicating the infiltration to deeper soil layers was impaired.

### 3.2 Plant development

RVI estimates are based on weekly measurements. Two temporal periods in particular were relevant for plant growth: i) the period from germination to the non-growing season in winter; and ii) the growing period after winter until harvest (Fig. A1). The maximum RVI values were all reached at a similar time (May 15, 2021 to May 18, 2021). In this regard, the non-eroded soil LV-cc n-d had the highest RVI (16.46 on average), while the non-eroded soil LV-cc d showed lower values (13.88 on average). The strongly eroded soil of LV-ng revealed the same pattern with a higher RVI for n-d (12 on average) and lower

RVI on d (10.35 on average) treatments. The extremely eroded soil of RG-ca, on the other hand, showed huge differences

between the n-d and d treatments (10.95 vs. 5.87 on average). Apart from that, the maximum standard deviation differed between n-d and d treatments for the three soil types (LV-cc: 1.65 < 3.29; LV-ng: 1.09 < 1.94; RG-ca: 1.17 < 0.82). Higher RVI values were already reached in non-eroded and strongly eroded soils compared to extremely eroded soil during the initial growing season in fall of 2020 until the non-growing season. Thus, mean RVI values of 4.47 to 6.63 were obtained for non-eroded and strongly eroded soils, while the extremely eroded soils had mean RVI values of only 3.61 (n-d) and 2.31 (d).

### 3.3 ET fluxes

The seasonal development (Fig. 3) of the quality-screened measured ET fluxes is similar for all treatments: a short growth phase after germination (1 - 2 mmol $m^{-2}$ $s^{-1}$) is followed by a decrease of fluxes until and during the non-growing season in winter (< 0.1 mmol $m^{-2}$ $s^{-1}$), when hardly any plant activity is found due to low temperatures. After the non-growing season, fluxes quickly return to their maximum fall level (1 - 2 mmol $m^{-2}$ $s^{-1}$) and then increase rapidly (> 5 mmol $m^{-2}$ $s^{-1}$). On the non-eroded soil (LV-cc), this rapid increase continued into June, while ET fluxes on the eroded soils (LV-ng and RZ-ca) already peaked in May. In addition, there is a clear difference in the maximum fluxes measured between soil types with 6.7 mmol $m^{-2}$ $s^{-1}$ for both treatments of non-eroded LV-cc, 5.6 / 6.5 mmol $m^{-2}$ $s^{-1}$ (n-d / d) for LV-ng, and 5.8 / 5.1 mmol $m^{-2}$ $s^{-1}$ (n-d / d) for RG-ca. Notably, there is a data gap from late April to late May due to sensor failure.

### 3.4 Modeling and validation

Pronounced differences of tested modeling approaches in terms of respective calibration statistics could be found. Calibration-model performances differ in their scatter around the 1:1-agreement plots (Fig. 4) and associated coefficients of determination (R2). NLR shows a clear overestimation at low and underestimation of higher ET fluxes (Table B4). Compared to that, MDV more accurately predicts low/high ET fluxes, but is characterized by a much lower precision due to a higher variance (Table B4). Among all modeling approaches, displayed calibration statistics (Table 2) indicate a very good or good (Table 1) prediction for SVM, ANN_BR, MDV and LUT over the entire range of observed ET fluxes. Considering the k-fold cross-validation (Fig. 5, Table 3), ANN_BR and SVM perform good, while MDV shows partially satisfactory statistics, and LUT shows unsatisfactory statistics due to allocation problems that arise when no class is found for the given climate conditions and the mean is used. Statistically, ANN_BR and SVM were similarly good in predicting observed ET fluxes (Table 2 and 3), even if they show a small tendency to overestimate low ET fluxes (Table B4). However, modelled ET fluxes using ANN_BR showed a large number of predicted negative ET fluxes (1547 on average per plot; Fig. A6) throughout the cultivation period. These fluxes occurred to an unrealistic degree during times when RH was significantly below saturation and plants were active (e.g., during the daytime period), resulting in a reduction in $ET_{sum}$ between 1 and 51 mm, depending on the treatment. This is most likely a method-specific extrapolation problem (see discussion) and the reason we use SVM for final budget estimations.

## 3.5 Diurnal ET fluxes, $ET_{sum}$ and crop ET

The model was able to predict the diurnal trends of ET fluxes during the cultivation period (Fig. 6). One representative day per month was selected in terms of the highest number of measurements. The two days of September and May have a reduced accuracy (ME: - 0.22 and – 0.3) due to a slight overestimation by the SVM modeling, while most of the other days are modeled accurately (ME <= ± 0.06). Additionally, the seasonal development of the SVM-predicted ET is depicted in Fig. 7 (see Fig. A3 - A5 for the other modeling approaches) and demonstrates a similar pattern to the measured fluxes described in Section 3.3.

In general, eroded soils tend to have a negative effect on $ET_{sum}$. However, this trend was not statistically significant (Kruskal-Wallis-Test, $ET_{sum}$: $\chi2 = 3.04$, df = 5, p = 0.69). For DM and WUE, on the other hand, the Kruskal-Wallis test indicated a significant difference between treatments (DM: $\chi^2 = 14.58$, df = 5, p = 0.01; WUE: $\chi^2 = 11.12$, df = 5, p = 0.05). The subsequent Dunn-Bonferroni post-hoc test revealed only a significant difference in DM between non-eroded LV-cc n-d and eroded RG-ca d (p = 0.013). However, no statistically significant pairwise differences were found for WUE. The amount of plant biomass in dry mass (DM) [kg] is decreasing from n-d to d and from less eroded soil types to more eroded soil types. DM ranges from $1.5 \pm 0.13$ kg m$^{-2}$ for LV-cc n-d to $0.85 \pm 0.2$ kg m$^{-2}$ for RG-ca d. $WUE_{agro}$ is decreasing from less eroded to more eroded soil types ranging from $7.25 \pm 1.23$ g DM kg$^{-1}$ H$_2$O to $4.69 \pm 0.71$ DM kg$^{-1}$ H$_2$O (Fig. 9).

In order to compare the individual treatments, daily ET and $ET_{sum}$ were calculated (Fig. 7). ET was affected by T, RH, PAR, and RVI, whereas only a small correlation was found with SM (Fig. 8). Higher ET fluxes were induced by increases in T, PAR and RVI, whereas increasing RH resulted in lower ET fluxes. $ET_{sum}$ (Fig. 9a) ranges between $212 \pm 45$ mm (LV-cc n-d) and $180 \pm 29$ mm (RG-ca d).

$ET_0$ for the observed study period (September 2020 – June 2021) and region (Uckermark) was 370 mm (DWD, 2022). We used the monthly values to calculate the $ET_c$ using $ET_0$ and the crop coefficient (Allen, 1998), resulting in an $ET_c$ of 263 mm for the cultivation period.

## 4 Discussion

In the following, we will discuss i) the effects of soil type and topsoil dilution on $ET_{sum}$, yield (DM) and $WUE_{agro}$, along with ii) the spatio-temporal variability of ET fluxes over the cultivation period, and iii) the suitability of the modeling strategies used in this study as well as potential ways forward to improve our approaches.

### 4.1 Effects of soil type and topsoil dilution on ET

In the studied region, soil types vary in their suitability for agricultural cultivation (MLUK, 2020). Luvisols support large water fluxes due to their clay-depleted, deep topsoils in combination with the clay-enriched and mostly decalcified subsoils. They are among the most productive soils in Brandenburg (MLUK, 2020; Stahr, 2022). Regosols are generally only moderately suitable for arable farming. They are usually found on hilltops and are characterized by parent material near to the surface,

lack of depth development, and limited rootability due to the dense, carbonate-rich parent material. They typically have low water availability and plant growth (Herbrich et al., 2018). They are formed by erosion of agricultural Luvisols as relatively organic matter rich top-soil is removed and deeper, nutrient-depleted lower soil layers are mixed into the cultivated layer (Arriaga and Lowery, 2003; Pimentel and Kounang, 1998).

The carried out topsoil dilution aimed at testing one of the processes of an approach to enhance soil C storage through topsoil deepening. Topsoil deepening through deeper ploughing might store originally topsoil-bound SOC in the deeper subsoil and generate SOC recharge in the diluted C poor topsoil (Sommer et al., 2016). The latter being tested during this study by the carried out topsoil dilution. However, side effects include, similar to erosion, nutrient deficiency and weaker rootability leading to decreased crop growth and yield (Al-Kaisi and Grote, 2007; Schneider et al., 2017; Feng et al., 2020). The boundary soil

conditions established by erosion and topsoil dilution may not only impact crop growth and yield but also disrupt the crop water balance, especially with the expected increase in drought and heat events in Central Europe (Spinoni et al., 2018). Consequently, farmers might become limited in their choice of crops due to water availability.

As predicted, we observed a significant decline in yield with erosion and topsoil dilution during the study period. However, the impact of soil-type-specific erosion intensity and topsoil dilution on $ET_{sum}$ was not as pronounced and the trend of declining

$ET_{sum}$ with soil type and top soil dilution was not statistically significant among all treatments ($212 \pm 45$ mm on non-eroded Calcic Luvisol to $180 \pm 29$ mm on extremely eroded topsoil diluted Calcaric Regosol). Notably, the studied year 2020/21 was comparatively wet (231.1 mm precipitation during the observed period), and potential effects of lower rootability and enhanced drought stress were not observed during the main growth period. This is of great importance because the Uckermark region generally has an overall water balance of about 1 (precipitation input equals $ET_{pot}$ output) and is therefore water or energy

limited depending on the annual precipitation and $ET_{pot}$ of each year. For example, the extremely dry year of 2018 was very likely water limited with an annual precipitation of $< 450$ mm and a predicted $ET_{pot}$ of $> 650$ mm and thus by far exceeding annual precipitation. However, the year 2021 had an annual precipitation of about 600 mm and a predicted $ET_{pot}$ of $< 575$ mm (DWD, 2022). Hence, in rather wet years, like the observed 2021, plant growth in the region is rather energy limited (of course dependent on precipitation during the growth period). This fits with our results, as during the studied period, most plots had a

lower $ET_{sum}$ than cumulative precipitation. However, it is very likely that the $ET_{pot}$/Pr ratio, and in fact also the observed actual $ET_{sum}$/Pr ratio will vary considerably between wetter and drier years and between different crops (particularly winter vs. summer crops).

Additionally, the observed imbalance of response in yield vs. $ET_{sum}$ led to significantly reduced $WUE_{agro}$. In a period of consecutive dry years, a lower $WUE_{agro}$ could additionally have a negative effect on the water and carbon balance of the whole

system, since the water consumption becomes less efficient, especially for the Calcaric Regosol (up to 36% less yield per used amount of water; Meena et al., 2020). This could further exacerbate the drought and potentially lead to legacy effects. Finally, winter crops and especially winter rye, are more resilient to drought (Ehlers, 1997) due to their head start in growth. Hence, a long-term investigation including other crops (e.g., summer cereal crops) and management strategies would be particularly interesting, as a greater decrease in ET may be observed with soil-specific erosion intensity.

## 4.2 Seasonal variability of ET fluxes and WUE

Over the entire cultivation period, ET fluxes responded particularly to crop growth, first during the establishment period in fall (mid of October to mid of November) and then again during the main growth period in spring (end of March to mid of May). The close relation of measured ET flux dynamics to RVI (Fig. 8; e.g., Hanks et al., 1969) can be associated with increasing T rates that strongly compensate for the suppression of E, as canopy biomass increases (Dubbert et al., 2014; Groh et al., 2020). Over the diurnal cycle, ET reacted to changes in environmental conditions, particularly temperature and RH, which together define the vapor-pressure deficit (VPD), as well as PAR. In particular, crops that have been bred to prioritize carbon gain over water conservation will tend to respond to rising VPD strongly (Dubbert et al., 2014; Massmann et al., 2019). Air temperature, humidity and PAR together with increasing biomass (expressed as higher RVI) controlled ET variability during the peak growth period in spring until harvest. SM, on the other hand, did not have a limiting effect on ET, which we attribute to the wet conditions during the observation period (see above), confirming that the observed crop cycle was not limited by water availability.

One of our expectations was that differences in $ET_{sum}$ would result mainly from differences during the main vegetation period from April to harvest due to variations in biomass and thus T. However, while the growing season between April and June is responsible for a large portion of $ET_{sum}$, ranging from 66 % to 73 %, it is only responsible for a small portion of differences between treatments, with a maximum of 14.3 mm from the non-eroded soil types. The combined fall and winter period, on the other hand, is responsible for a difference of up to 17.5 mm in $ET_{sum}$ between non-eroded and extremely eroded soil types, although it accounts for 27 % to 34 % of $ET_{sum}$ only. This is interesting, because it suggests that the reason behind the soil type differences in ET for winter rye are caused by static differences (e.g., lower biomass) and suppressed E (e.g., a shift in the T/ET ratio) rather than dynamic differences (e.g., the vegetation responses to environmental drivers or drought). This should be further evaluated by partitioning ET into T and E. The described FluxCrane is particularly suited for such an approach by combining flux and in-situ stable isotope approaches (Dubbert et al., 2014; Rothfuss et al., 2021). Beside the overall slight reduction of $ET_{sum}$ on eroded soil types and topsoil diluted treatments, measured ET fluxes were larger on extremely eroded plots at the beginning of the growing season before canopy closure which could be explained by a lower soil cover. This may be related to the fact that a lower vegetation cover, which is visible in the RVI, can lead to higher E prior to canopy closure (Dubbert et al., 2014; Hu et al., 2009; Raz-Yaseef et al., 2012; Wang et al., 2012).

## 4.3 Modeling approaches

Methodologically, the study faced two main challenges: accurately quantifying $ET_{sum}$ and realistically predicting diurnal variations during both, the low ET winter and high ET summer periods. Among the modeling approaches compared in this study, only NLR showed calibration statistics less than good (Table 2). While the LUT showed very good calibration results, the allocation problems that occur when no class is found (Fig. 5) and the mean is used, resulted in the lowest predictive ability during validation over the full range of measured ET fluxes. Some studies also obtained quite plausible results for LUT and

MDV (Boudhina et al., 2018; Falge et al., 2001a; Moffat et al., 2007), and adjusting the classes of the LUT could further improve the results of this approach. However, with the available dataset, the only way to avoid allocation problems was to use fewer classes. This resulted in a loss of variability, making diurnal differences invisible and ET estimates less accurate.

MDV, on the other hand, partially showed only satisfactory values during validation (Table 3), while SVM and ANN performed good or very good according to the defined classes (Table 1). Additionally, previous studies found that MDV (as well as LUT) performs particularly weakly with increasingly large gaps (Moffat et al., 2007; Kim et al., 2020). Especially for conditions where no measurements could take place due inter-alia environmental conditions (large gaps in winter), the fact that MDV takes averaged values of adjacent measurements could explain the rather bad predictions. This is similar for LUT,

since no classes could be created for conditions where no measurements took place. The machine learning approaches SVM and ANN_BR, on the other hand, are not as sensitive to larger observational gaps because their training includes all measurements. For seasonal variability and budgets, we achieved the best performance with the SVM approach, while ANN showed reduced daily and seasonal cumulative ET due to an unrealistic amount of predicted negative fluxes (up to 51 mm; Fig. A5). However, the best approach for modeling or gap-filling can vary depending on the application and investigated

parameters. For example, in gap-filling methane fluxes using eddy covariance (Kim et al., 2020), ANN_BR was superior to SVM.

Another important aspect of modeling is potential over- or underestimation of fluxes. Shrestha and Shukla (2015), for example, attempted to predict actual lysimeter ET using different approaches (e.g., ANN_BR and SVM) and crops (pepper, watermelon) in a subtropical environment. Best predictions were obtained with SVM (pepper: 204.7 mm lysimeter vs. 181.8 mm SVM;

watermelon: 231.71 mm lysimeter vs. 189.83 mm SVM). However, they reported overestimation of low fluxes and underestimation of high fluxes by ANN_BR and SVM. In our study, we observed a tendency to slightly overestimate small fluxes using SVM based modeling (Fig. B4). In this regard, using plot specific multi-depth SM data could also improve the predicted $ET_{sum}$ based on SVM in the future. Similarly, we expected considering wind speed to improve ET prediction, but could not find an effect on observed ET for the study period.

Furthermore, it must be noted that the quality of SVM (and ANN_BR) predictions is highly dependent on the amount of data available (Chia et al. 2020; Abudu et al. 2010). Consequently, we tested the minimum amount of data necessary to provide predicted ET fluxes of good quality (see criteria in M&M). For the specific dataset, even as little as 50 % of the total data available (minimum 300 measurements) provided good results. Thus, we emphasize that capturing a large variability of fluxes under different environmental conditions seems to be more important than a merely large data set.

**4.4 Evaluation of the new system and comparison with other measurement techniques**

$ET_c$ was 263 mm during the cultivation period, which is comparable to our observed results ($ET_{sum}$) of 212 mm for non-eroded Calcic Luvisol. However, it is important to consider that $ET_0$ calculations using the Penman-Monteith equation (FAO56-PM) are reported to overestimate $ET_0$ and consequently $ET_c$ (Allen, 1998). Thus, our FluxCrane $ET_{sum}$ seem sensible overall. Nevertheless, it is advisable to directly compare them with lysimeter, and potentially drone-based observations of ET. This is

particularly relevant in light of ongoing discussions surrounding method constraints of estimating ET across scales (Ding et al., 2021; Ghiat et al., 2021; Hamel et al., 2015).

For instance, there is a nearby lysimeter experiment with multi treatment conducted by (Groh et al., 2020). They report a wide range of $ET_{sum}$ for the period between 2014 and 2018 (300 to 600 mm), with the lower range boundary being comparable to our results (considering that we only calculated budgets for the 9-months cultivation period and excluded the fallow period during the summer months with high ET). It has to be noted, however, that not only environmental conditions but also crops studied in Groh et al. (2020) varied from year to year and, more importantly, from our study, hampering comparability between studies. However, the direct vicinity of two large-scale setups able to estimate $ET_{sum}$ should be utilized in the future. Another lysimeter based study conducted in Brunswick (Lower Saxony, Germany) for a cultivation season of winter rye report a range of observed ET fluxes very comparable to our study, with less than 1 mm day$^{-1}$ in winter to a maximum of 6 - 7 mm day$^{-1}$ in summer (Bundesanstalt für Gewässerkunde, 2023).

Finally, modeling using statistical and empirical approaches is used in many fields e.g., to calculate of reference ET ($ET_0$) with limited meteorological parameters (Chia et al., 2020) or ET from eddy-covariance measurements as well as canopy chamber measurements (Hui et al., 2004; Moffat et al., 2007; Falge et al., 2001a; Falge et al., 2001b; Hamel et al., 2015; Kübert et al., 2019). The connection between modeling approaches in combination with the described continuous high-resolution long-term ET measurements of numerous small-scale treatments, gives additional opportunities to observe the progression of ET over an entire cultivation season and, for example, to identify key periods that drive overall treatment differences. Here, our combined approach of automated chamber measurements of ET with data driven modeling fills a unique application niche among the different methods to quantify ET fluxes. In comparison, eddy-covariance (e.g., Boudhina et al., 2018; Simpson et al., 2019), and lysimeter-based observations (Groh et al., 2020) are unparalleled in measurement frequency and therefore only require filling gaps, which are typically much smaller compared to our approach. However, eddy-covariance systems operating at the ecosystem scale are not able to detect small-scale spatial heterogeneities in ET fluxes, e.g., to test the effects of soil type, management, etc., or different crops grown simultaneously (Anapalli et al., 2018). Lysimeter approaches, on the other hand, are useful for combining small-scale spatial heterogeneity with high-frequency measurements, but are limited to water cycle applications and a direct link to carbon/GHG dynamics is not straightforward.

Classic manual gas exchange chamber applications are capable of analyzing small-scale spatial effects (e.g Macagga et al., 2023; Antonijević et al., 2023). However, despite the ability to observe diurnal cycles, the total amount of measured data is usually very limited. Typically, campaign-based measurements are performed approximately every 3 weeks (Huth et al., 2017). This results in a total maximum amount of about 300 fluxes per treatment (3 replicates, 6 measurements per plot per campaign). Even when measurement campaigns are performed more frequently, the available fluxes remaining after quality checks are quite limited (see e.g., Dubbert et al., 2014 with 22 measurement campaigns in 8 months resulting in ~297 fluxes per treatment). Under these conditions, combining chamber measurements with data-driven ET flux modeling approaches is usually limited to very simple approaches (e.g., Falge et al., 2001a). In the present study, the automated FluxCrane generated approximately 7-10 times the amount of ET fluxes compared to manually operated chambers. In addition, the system is not as

disruptive to plant growth. For example, permanently installed automated canopy chambers or manually conducted approaches, tend to physically harm the canopy and have condensation issues due to permanently installed tubing and inappropriate air mixing within the chamber (e.g., Hamel et al., 2015). Moreover, the ability to observe nighttime fluxes has great potential to study previously overlooked short-term dynamics in ET and to improve the representation of underlying processes in process-based hydrological modeling, compared to manually operated chambers. This offers several benefits: 1) dynamic developments in ET fluxes and differences between treatments are easier to analyze (even if only the measured fluxes are considered), 2) the much larger number of fluxes available bears the potential to apply data-driven ET flux modeling using advanced statistical, empirical, and machine-learning based algorithms.

## 4.4 Conclusion and outlook

We present a possibility to obtain plausible $ET_{sum}$ and diurnal cycles of ET by using the novel FluxCrane system in combination with data-driven SVM based modeling. We expected strong negative effects of eroded soils and topsoil dilution on $ET_{sum}$ as well as yield. However, crop yield responded much more strongly to eroded soils and topsoil dilution than $ET_{sum}$ in the observed rather wet year, leading to strong negative shifts in $WUE_{agro}$. The novel FluxCrane, combined with data driven modeling, fills the unique application to observe temporal flux dynamics and seasonal budgets for distinct landscape elements simultaneously. Thus, the new system has a large potential to bring new insights into water-flux dynamics and budgets. In combination with $CO_2$ measurements, the novel FluxCrane could give new insights in ecosystem WUE in a high spatio-temporal resolution using NEE (net ecosystem exchange). In addition, coupled with the GEP (gross ecosystem production) and innovative measurements such as in-situ stable water isotopes (Dubbert et al., 2014; Kübert et al., 2020), a separation of ET into T and E would be possible to assess crop performance by assessing plant specific WUE (Tallec et al., 2013) or to study root water uptake dynamics (Deseano Diaz et al., 2023; Kühnhammer et al., 2020). This is particularly relevant for regions with strong spatial heterogeneity in soils and generally low precipitation like the Uckermark and of crucial importance for the terrestrial water balance as well as the prediction of future ecosystem feedbacks (Groh et al., 2020).

## 5 Code availability

The codes produced for this study are available from the corresponding author upon request.

## 6 Data availability

The data sets produced for this study are available from the corresponding author upon request.

**7 Author contributions**

AD, MD conceived and planned the study. MS, GV, MH supervised automated measurements and conducted complimentary field measurements. AD analyzed the data and wrote a first version of the manuscript. All authors contributed to manuscript writing.

**8 Competing interests**

The authors declare that they have no conflict of interest.

**9 Acknowledgements**

The authors acknowledge funding by the German Federal Ministry of Food and Agriculture (FNR Grant: 22404117) and the German Science Foundation (DFG Grant DU1688/6-1). The authors are very grateful to the Pfannenstiel ProProject GmbH
for the excellent collaboration in designing as well as constructing the gantry crane system and ATTEC Automation GmbH for programming the system control. The study was a part of the CarboZALF project of the Leibniz Centre for Agricultural Landscape Research (ZALF). The CarboZALF project was financially supported by the German Federal Ministry of Food and Agriculture (BMEL) and the Ministry of Environment, Health and Climate Protection (MLUK) of the State of Brandenburg. The authors thank the Experimental Infrastructure Platform (EIP) of ZALF for assistance with field measurements and John
Marshal and Sebastian Fitting for proofreading the manuscript.




**Publication bibliography**

Abudu, S; B., A.S.; King, J.P.: Infilling Missing Daily Evapotranspiration Data Using Neural Networks, J. Irrig. Drain Eng. 136 (5), 317–325, https://doi.org/10.1061/(ASCE)IR.1943-4774.0000197, 2010.

Al-Kaisi, M.M.; Grote, J.B.: Cropping Systems Effects on Improving Soil Carbon Stocks of Exposed Subsoil, Soil Sci. Soc. Am. J. 71 (4), 1381–1388, https://doi.org/10.2136/sssaj2006.0200, 2007.

Allen, R.G.; Pereira, L.S., Raes, D.; Smith, M. (Eds.): Crop evapotranspiration: Guidelines for computing crop water requirements, Irrig. Drain. Paper 56, FAO, Rome, Italy, 300 pp., 92-5-104219-5, 1998.

Amt für Statistik Berlin-Brandenburg: Ernteberichterstattung über Feldfrüchte und Grünland im Land Brandenburg 2019,
https://www.statistik-berlin-brandenburg.de/, last access: 18 August 2022.

Anapalli, S.S.; Fisher, D.K.; Reddy, K.N.; Wagle, P.; Gowda, P.H.; Sui, R.: Quantifying soybean evapotranspiration using an eddy covariance approach, Agr. Water Manage. 209, 228–239, https://doi.org/10.1016/j.agwat.2018.07.023, 2018.

Antonijević, D.; Hoffmann, M.; Prochnow, A.; Krabbe, K.; Weituschat, M.; Couwenberg, J.; Ehlert, S.; Augustin, J.: The unexpected long period of elevated $CH_4$ emissions from an inundated fen meadow ended only with the occurrence of cattail
(Typha latifolia), Glob. Change Boil. 29 (13), 3678–3691, https://doi.org/10.1111/gcb.16713, 2023.

Arriaga, F.J; Lowery, B.: Corn production on an eroded soil: effects of total rainfall and soil water storage, Soil Till. Res. 71 (1), 87–93, https://doi.org/10.1016/S0167-1987(03)00040-0, 2003.

Bakker, M.M.; Govers, G.; Jones, R.A.; Rounsevell, M.D.A.: The Effect of Soil Erosion on Europe's Crop Yields, Ecosystems 10 (7), 1209–1219, https://doi.org/10.1007/s10021-007-9090-3, 2007.

Bundesanstalt für Gewässerkunde: 2.12 Mean Annual Potential Evaporation Depth as Grass Reference Evapotranspiration, https://geoportal.bafg.de/dokumente/had/212GrassReferenceEvapotraspiration.pdf, last access: 10 March 2023.

Den Biggelaar, C.; Lal, R.; Wiebe, K.; Breneman, V.: The Global Impact Of Soil Erosion On Productivity: I: Absolute and Relative Erosion-induced Yield Losses, Adv. Agron. 81, 1–48, 2003.

Bishop, C.M.: Neural networks for pattern recognition, Clarendon Press, Oxford, 482 pp., 1995.

Blum, W.E.H.: Soil and Land Resources for Agricultural Production: General Trends and Future Scenarios-A Worldwide Perspective, International Soil and Water Conservation Research 1 (3), 1–14, https://doi.org/10.1016/S2095-6339(15)30026-5, 2013.

Boudhina, N.; Zitouna-Chebbi, R.; Mekki, I.; Jacob, F.; Ben Mechlia, N.; Masmoudi, M.; Prévot, L.: Evaluating four gap-filling methods for eddy covariance measurements of evapotranspiration over hilly crop fields, Geosci. Instrum. Meth. 7
(2), 151–167, https://doi.org/10.5194/gi-7-151-2018, 2018.

Chia, M.Y.; Huang, Y.F.; Koo, C.H.: Support vector machine enhanced empirical reference evapotranspiration estimation with limited meteorological parameters, Comput. Electron. Agr. 175, 105577, https://doi.org/10.1016/j.compag.2020.105577, 2020.

Deseano Diaz, P.A.; van Dusschoten, D.; Hubert, A.; Brüggemann, N.; Javaux, M.; Merz, S.; Vanderborght, J.; Vereecken, H.; Dubbert, M.; Rothfuss, Y.: Response of a grassland species to dry environmental conditions from water stable isotopic monitoring: no evident shift in root water uptake to wetter soil layers, Plant Soil 482 (1-2), 491–512, https://doi.org/10.1007/s11104-022-05703-y, 2023.

Ding, J.; Li, S.; Wang, H.; Wang, C.; Zhang, Y.; Yang, D.: Estimation of Evapotranspiration and Crop Coefficient of Chinese Cabbage Using Eddy Covariance in Northwest China, Water-SUI 13 (19), 2781, https://doi.org/10.3390/w13192781, 2021.

Doetterl, S.; Berhe, A.A.; Naidu, E.; Wang, Z.; Sommer, M.; Fiener, P.: Erosion, deposition and soil carbon: A review of process-level controls, experimental tools and models to address C cycling in dynamic landscapes, Earth-Sci. Rev. 154, 102–122, https://doi.org/10.1016/j.earscirev.2015.12.005, 2016.

Dubbert, M.; Piayda, A.; Cuntz, M.; Correia, A.C.; Costa e Silva, F.; Pereira, J.S.; Werner, C.: Stable oxygen isotope and flux partitioning demonstrates understory of an oak savanna contributes up to half of ecosystem carbon and water exchange, Front. Plant Sci. 5, 530, https://doi.org/10.3389/fpls.2014.00530, 2014.

DWD: Klimareport Brandenburg. 1. Auflage, German Weather Service, Germany, Offenbach am Main, 40 pp., 2019.

DWD: Climatological maps of Germany, https://www.dwd.de/EN/ourservices/klimakartendeutschland/klimakartendeutschland.html, last access: 10 March 2023.

Ehlers, W.: Zum Transpirationskoeffizienten von Kulturpflanzen unter Feldbedingungen. Pflanzenbauwissenschaften 1 (3), 97–108, 1997.

Falge, E.; Baldocchi, D.; Olson, R.; Anthoni, P.; Aubinet, M.; Bernhofer, C.; Burba, G.; Ceulemans, R.; Clement, R.; Dolman, H.; Granier, A.; Gross, P.; Grünwald, T.; Hollinger, D.; Jensen, N.O.; Katul, G.; Keronen, P.; Kowalski, A.; Lai, C.T.; Law, B.E.; Meyers, T.; Moncrieff, J.; Moors, E.; Munger, J.W.; Pilegaard, K.; Rannik, Ü.; Rebmann, C.; Suyker, A.E.; Tenhunen, J.; Tu, K.; Verma, S.; Vesala, T.; Wilson, K.; Wofsy, S.: Gap filling strategies for defensible annual sums of net ecosystem exchange, Agr. Forest Meteorol. 107 (1), 43–69, https://doi.org/10.1016/S0168-1923(00)00225-2, 2001a.

Falge, E.; Baldocchi, D.; Olson, R.; Anthoni, P.; Aubinet, M.; Bernhofer, C.; Burba, G.; Ceulemans, R.; Clement, R.; Dolman, H.; Granier, A.; Gross, P.; Grünwald, T.; Hollinger, D.; Jensen, N.O.; Katul, G.; Keronen, P.; Kowalski, A.; Lai, C.T.; Law, B.E.; Meyers, T.; Moncrieff, J.; Moors, E.; Munger, J.W.; Pilegaard, K.; Rannik, Ü.; Rebmann, C.; Suyker, A.E.; Tenhunen, J.; Tu, K.; Verma, S.; Vesala, T.; Wilson, K.; Wofsy, S.: Gap filling strategies for long term energy flux data sets, Agr. Forest Meteorol. 107 (1), 71–77, https://doi.org/10.1016/S0168-1923(00)00235-5, 2001b.

Feng, Q.; An, C.; Chen, Z.; Wang, Z.: Can deep tillage enhance carbon sequestration in soils? A meta-analysis towards GHG mitigation and sustainable agricultural management. In Renewable and Sustainable Energy Reviews 133, 110293, https://doi.org/10.1016/j.rser.2020.110293, 2020.

Fohrer, N.; Bormann, H.; Miegel, K.; Casper, M.; Schumann, A.; Bronstert, A.; Weiler, M.: Hydrologie, 1. Edition, Haupt Verlag (Utb Basics, 4513), Bern, Switzerland, 2016.

Ghiat, I.; Mackey, H.R.; Al-Ansari, T.: A Review of Evapotranspiration Measurement Models, Techniques and Methods for Open and Closed Agricultural Field Applications, Water 13 (18), 2523, https://doi.org/10.3390/w13182523, 2021.

Görres, C.-M.; Kutzbach, L.; Elsgaard, L.: Comparative modeling of annual CO2 flux of temperate peat soils under permanent grassland management, Agr. Ecosyst. Environ. 186, 64–76, https://doi.org/10.1016/j.agee.2014.01.014, 2014.

Groh, J.; Diamantopoulos, E.; Duan, X.; Ewert, F.; Herbst, M.; Holbak, M.; Bahareh Kamali, B.; Kersebaum, K.-C.; Kuhnert, M.; Lischeid, G.; Nendel, C.; Priesack, E.; Steidl, J.; Sommer, M.; Pütz. T.; Vereecken, H.; Wallor E.; Weber, T.K.D.; Wegehenkel, M.; Weihermüller L.; Gerke, H.H.: Crop growth and soil water fluxes at erosion-affected arable sites: Using weighing lysimeter data for model intercomparison, Vadose Zone J. 19 (1), https://doi.org/10.1002/vzj2.20058, 2020.

Hagan, M.T.; Demuth, H.B.; Beale, M.; De Jesus, O.: Neural Network Design, 1. Edition, PWS Pub. Co., Boston, USA, 800
pp., 978-0971732117, 1996.

Hamel, P.; Mchugh, I.; Coutts, A.; Daly, E.; Beringer, J.; Fletcher, T.D.: Automated Chamber System to Measure Field Evapotranspiration Rates, J. Hydrol. Eng. 20 (2), 04014037, https://doi.org/10.1061/(ASCE)HE.1943-5584.0001006, 2015.

Hanson, R.L.: Evapotranspiration and droughts, US Geological Survey Water-Supply Paper, USA, National Water Summary
1988-89 2375, Hydrologic Events and Floods and Droughts, 99–104, 1991.

Hatfield, J.L.; Dold, C.: Water-Use Efficiency: Advances and Challenges in a Changing Climate, Front. Plant Sci. 10, 103. DOI: https://doi.org/10.3389/fpls.2019.00103, 2019.

Herbrich, M.; Gerke, H.H.; Sommer, M.: Root development of winter wheat in erosion-affected soils depending on the position in a hummocky ground moraine soil landscape, J. Plant Nutr. Soil Sc. 181 (2), 147–157,
https://doi.org/10.1002/jpln.201600536, 2018.

Hoffmann, M.; Jurisch, N.; Albiac Borraz, E.; Hagemann, U.; Drösler, M.; Sommer, M.; Augustin, J.: Automated modeling of ecosystem CO 2 fluxes based on periodic closed chamber measurements: A standardized conceptual and practical approach, Agr. Forest Meteorol.  200, 30–45, https://doi.org/10.1016/J.AGRFORMET.2014.09.005, 2015.

Hu, Z.; Yu, G.; Zhou, Y.; Sun, X.; Li, Y.; Shi, P.; Wang, Y.; Song, X.; Zheng, Z.; Zhang, L.; Li, S.: Partitioning of
evapotranspiration and its controls in four grassland ecosystems: Application of a two-source model, Agr. Forest Meteorol. 149 (9), 1410–1420, https://doi.org/10.1016/J.AGRFORMET.2009.03.014, 2009

Hui, D.; Wan, S.; Su, B.; Katul, G.; Monson, R.; Luo, Y.: Gap-filling missing data in eddy covariance measurements using multiple imputation (MI) for annual estimations. Agr. Forest Meteorol. 121 (1-2), 93–111, https://doi.org/10.1016/S0168-1923(03)00158-8, 2004.

Huth, V.; Vaidya, S.; Hoffmann, M.; Jurisch, N.; Günther, A.; Gundlach, L.; Hagemann, U.; Elsgaard, L.; Augustin, J.: Divergent NEE balances from manual-chamber $CO_2$ fluxes linked to different measurement and gap-filling strategies: A source for uncertainty of estimated terrestrial C sources and sinks?, J. Plant Nutr. Soil Sc 180 (3), 302–315, https://doi.org/10.1002/jpln.201600493, 2017.

Ichii, K.; Ueyama, M.; Kondo, M.; Saigusa, N.; Kim, J.; Alberto, Ma.C., Ardö, J.; Euskirchen, E.S.;  Kang, M.; Hirano, T.;
Joiner, J.; Kobayashi, H.; Marchesini, L.B.; Merbold, L.; Miyata, A.; Saitoh, T.M.; Takagi, K.; Varlagin, A.; Bret-Harte, M.S.;  Kitamura, K.; Kosugi, Y.; Kotani, A.; Kumar, K.; Li, S.G.; Machimura, T.; Matsuura, Y.; Mizoguchi, Y.; Ohta, T.;

Mukherjee, S.; Yanagi, Y.; Yasuda, Y.; Zhang, Y.; Zhao F.: New data-driven estimation of terrestrial $CO_2$ fluxes in Asia using a standardized database of eddy covariance measurements, remote sensing data, and support vector regression, J. Geophys. Res-Biogeo 122 (4), 767–795, https://doi.org/10.1002/2016JG003640, 2017.

Jie, C.; Jing-zhang, C.; Man-zhi, T.; Zi-tong, G.: Soil degradation: a global problem endangering sustainable development, J. Geogr. Sci. 12 (2), 243–252, https://doi.org/10.1007/BF02837480, 2002.

Kandel, T.P.; Elsgaard, L.; Laerke, P.E.: Measurement and modelling of $CO_2$ flux from a drained fen peatland cultivated with reed canary grass and spring barley, GCB Bioenergy 5 (5), 548–561, https://doi.org/10.1111/gcbb.12020, 2013.

Katerji, N.; Mastrorilli, M.; Rana, G.: Water use efficiency of crops cultivated in the Mediterranean region: Review and 680 analysis, Eur. J. Agron. 28 (4), pp. 493–507, https://doi.org/10.1016/j.eja.2007.12.003, 2008.

Kim, Y.; Johnson, M.S.; Knox, S.H.; Black, T.A.; Dalmagro, H.J.; Kang, M.; Kim, J.; Baldocchi, D.: Gap-filling approaches for eddy covariance methane fluxes: A comparison of three machine learning algorithms and a traditional method with principal component analysis, Glob. Change Biol. 26 (3), 1499–1518, https://doi.org/10.1111/GCB.14845, 2020.

Kişi, O.; Çimen, M.: Evapotranspiration modelling using support vector machines / Modélisation de l'évapotranspiration à 685 l'aide de 'support vector machines', Hydrolog. Sci. J. 54 (5), 918–928, https://doi.org/10.1623/hysj.54.5.918, 2009.

Kubat, M.: Neural networks: a comprehensive foundation by Simon Haykin, Macmillan, 1994, ISBN 0-02-352781-7, Knowl. Eng. Rev. 13 (4), 409–412, https://doi.org/10.1017/S0269888998214044, 1999.

Kübert, A.; Götz, M.; Kuester, E.; Piayda, A.; Werner, C.; Rothfuss, Y.; Dubbert, M.: Nitrogen Loading Enhances Stress Impact of Drought on a Semi-natural Temperate Grassland, Front. Plant Sci. 10, 1051, 690 https://doi.org/10.3389/fpls.2019.01051, 2019.

Kübert, A.; Paulus, S.; Dahlmann, A.; Werner, C.; Rothfuss, Y.; Orlowski, N.; Dubbert, M.: Water Stable Isotopes in Ecohydrological Field Research: Comparison Between In Situ and Destructive Monitoring Methods to Determine Soil Water Isotopic Signatures, Front. Plant Sci. 11, 387, https://doi.org/10.3389/fpls.2020.00387, 2020.

Kühnhammer, K.; Kübert, A.; Brüggemann, N.; Deseano Diaz, P.; van Dusschoten, D.; Javaux, M.; Merz, S.; Vereecken, H.; 695 Dubbert, M.; Rothfuss, Y.: Investigating the root plasticity response of Centaurea jacea to soil water availability changes from isotopic analysis, New Phytol. 226 (1), 98–110, https://doi.org/10.1111/nph.16352, 2020.

Livingston, G.P.; Hutchinson, G.L.: Enclosure-based measurement of trace gas exchange: applications and sources of error, Biogenic trace gases: measuring emissions from soil and water 51, 14–51, Oxford, England, 1995.

Louwagie, G.; Gay, S.H.; Sammeth, F.; Ratinger, T.: The potential of European Union policies to address soil degradation in 700 agriculture, Land Degrad. Dev. 22 (1), 5–17, https://doi.org/10.1002/ldr.1028, 2011.

Macagga, R.; Asante, M.; Sossa, G.; Antonijevic, D.; Dubbert, M.; Hoffmann, M.: Validation and field application of a low-cost device to measure $CO_2$ and ET fluxes, EGUsphere [preprint], https://doi.org/10.5194/egusphere-2023-553, 2023.

Massmann, A.; Gentine, P.; Lin, C.: When Does Vapor Pressure Deficit Drive or Reduce Evapotranspiration?, J. Adv. Model. Earth Sys. 11 (10), 3305–3320, https://doi.org/10.1029/2019MS001790, 2019.

Meena, R.S.; Kumar, S.; Yadav, G.S.: Soil Carbon Sequestration in Crop Production, in: Nutrient Dynamics for Sustainable Crop Production, edited by: Meena, R.S., Springer, Singapore, 1–39, https://doi.org/10.1007/978-981-13-8660-2, 2020.

MLUK: Steckbriefe Brandenburger Böden. Sammelmappe, Ministerium für Ländliche Entwicklung, 2020.

Moffat, A.M.; Papale, D.; Reichstein, M.s; Hollinger, D.Y.; Richardson, A.D.; Barr, A.G.; Beckstein, C.; Braswell, B.H.; Churkina, G.; Desai, A.R.; Falge, E.; Gove, J.H.; Heimann, M.; Hui, D.; Jarvis, A.J.; Kattge, J.; Noormets, A.; Stauch, V.J.: Comprehensive comparison of gap-filling techniques for eddy covariance net carbon fluxes, Agr. Forest Meteorol. 147 (3-4), 209–232, https://doi.org/10.1016/j.agrformet.2007.08.011, 2007.

Moriasi, D.N.; Gitau, M.W.; Prasad Daggupati, N.Pai.: Hydrologic and Water Quality Models: Performance Measures and Evaluation Criteria, T. ASABE 58 (6), 1763–1785, https://doi.org/10.13031/TRANS.58.10715, 2015.

Noble, W.S.: What is a support vector machine? Nat. Biotechnol. 24 (12), 1565–1567. https://doi.org/10.1038/nbt1206-15652006, 2006.

Öttl, L.K.; Wilken, F.; Auerswald, K.; Sommer, M.; Wehrhan, M.; Fiener, P.: Tillage erosion as an important driver of in-field biomass patterns in an intensively used hummocky landscape, Land Degrad Dev. 32 (10), 3077–3091, https://doi.org/10.1002/LDR.3968, 2021.

Pape, L.; Ammann, C.; Nyfeler-Brunner, A.; Spirig, C.; Hens, K.; Meixner, F. X.: An automated dynamic chamber system for surface exchange measurement of non-reactive and reactive trace gases of grassland ecosystems, Biogeosciences 6 (3), 405–429, https://doi.org/10.5194/bg-6-405-2009, 2009.

Pimentel, D.; Kounang, N.: Ecology of Soil Erosion in Ecosystems. In Ecosystems 1 (5), 416–426, https://doi.org/10.1007/s100219900035, 1998.

Raz-Yaseef, N.; Yakir, D.; Schiller, G.; Cohen, S.: Dynamics of evapotranspiration partitioning in a semi-arid forest as affected by temporal rainfall patterns, Agr. Forest Meteorol. 157, 77–85, https://doi.org/10.1016/J.AGRFORMET.2012.01.015, 2012.

Rojas, R. (Eds.): Neural Networks, Springer, Berlin, Heidelberg, Germany, 1996.

Rothfuss, Y.; Quade, M.; Brüggemann, N.; Graf, A.; Vereecken, H.; Dubbert, M.: Reviews and syntheses: Gaining insights into evapotranspiration partitioning with novel isotopic monitoring methods, Biogeosciences 18 (12), 3701–3732, https://doi.org/10.5194/bg-18-3701-2021, 2021.

Schad, P.: The international soil classification system WRB, 2014, in: Novel methods for monitoring and managing land and water resources in Siberia, edited by: Mueller, L., Sheudshen, A. K., & Eulenstein, F., Springer International Publishing, 563-571, 2016.

Schappert, S.: Wie wird Niederschlag beim DWD gemessen und wo fällt am meisten? https://www.dwd.de/DE/wetter/thema_des_tages/2018/11/28.html, last access 18 August 2022.

Schmidt, A.; Creason, W.; Law, B.E.: Estimating regional effects of climate change and altered land use on biosphere carbon fluxes using distributed time delay neural networks with Bayesian regularized learning, Neural Networks 108, 97–113, https://doi.org/10.1016/j.neunet.2018.08.004, 2018.

Schneider, F.; Don, A.: Root-restricting layers in German agricultural soils. Part I: extent and cause, Plant Soil 442 (1-2), 433–451, https://doi.org/10.1007/s11104-019-04185-9, 2019.

Schneider, F.; Don, A.; Hennings, I.; Schmittmann, O.; Seidel, S.J.: The effect of deep tillage on crop yield – What do we really know?, Soil Till. Research 174, 193–204, https://doi.org/10.1016/j.still.2017.07.005, 2017.

Searchinger, T.D.; Wirsenius, S.; Beringer, T.; Dumas, P.: Assessing the efficiency of changes in land use for mitigating climate change, Nature 564 (7735), 249–253, https://doi.org/10.1038/s41586-018-0757-z, 2018.

Shrestha, N.K.; Shukla, S.: Support vector machine based modeling of evapotranspiration using hydro-climatic variables in a sub-tropical environment, Agr. Forest Meteorol. 200, 172–184, https://doi.org/10.1016/j.agrformet.2014.09.025, 2015.

Simpson, G.; Runkle, B.R.K.; Eckhardt, T.; Kutzbach, L.: Evaluating closed chamber evapotranspiration estimates against eddy covariance measurements in an arctic wetland, J. Hydrol. 578, 124030, https://doi.org/10.1016/j.jhydrol.2019.124030, 2019.

Sommer, M.; Augustin, J.; Kleber, M.: Feedbacks of soil erosion on SOC patterns and carbon dynamics in agricultural landscapes—The CarboZALF experiment, Soil Till. Res. 156 (156), 182–184, https://doi.org/10.1016/j.still.2015.09.015 2016.

Sommer, M.; Gerke, H.H.; Deumlich, D.: Modelling soil landscape genesis — A "time split" approach for hummocky agricultural landscapes. Geoderma 145 (3-4), 480–493, https://doi.org/10.1016/j.geoderma.2008.01.012, 2008.

Spinoni, J.; Vogt, J.V.; Naumann, G.; Barbosa, P.; Dosio, A.: Will drought events become more frequent and severe in Europe?, Int. J. Climatol 38 (4), 1718–1736, https://doi.org/10.1002/joc.5291, 2018.

Stahr, A.: Bodentypen, http://www.ahabc.de/bodentypen/, last access 19 August 2022.

Sturges, H. A.: The choice of a class interval, J. Am. Stat. Assoc. 21 (153), 65–66, https://doi.org/10.1080/01621459.1926.10502161, 1926.

Tallec, T.; Béziat, P.; Jarosz, N.; Rivalland, V.; Ceschia, E.: Crops' water use efficiencies in temperate climate: Comparison of stand, ecosystem and agronomical approaches, Agr. Forest Meteorol. 168, 69–81, https://doi.org/10.1016/j.agrformet.2012.07.008, 2013.

Trenberth, K.E.: Changes in precipitation with climate change, Clim. Res. 47 (1), 123–138, https://doi.org/10.3354/cr00953, 2011.

Vaidya, S.; Schmidt, M.; Rakowski, P.; Bonk, N.; Verch, G.; Augustin, J.; Sommer, M.; Hoffmann, M.: A novel robotic chamber system allowing to accurately and precisely determining spatio-temporal $CO_2$ flux dynamics of heterogeneous croplands, Agr. Forest Meteorol. 296, 108206, https://doi.org/10.1016/j.agrformet.2020.108206, 2021.

Wang, L.; D'Odorico, P.; Evans, J.P.; Eldridge, D.J.; McCabe, M.F.; Caylor, K.K.; King, E.G.: Dryland ecohydrology and climate change: critical issues and technical advances, Hydrol. Earth Syst. Sc. 16 (8), 2585–2603, https://doi.org/10.5194/HESS-16-2585-2012, 2012.

Wehrhan, M.; Rauneker, P.; Sommer, M.: UAV-Based Estimation of Carbon Exports from Heterogeneous Soil Landscapes--A Case Study from the CarboZALF Experimental Area, Sensors 16 (2), 255, https://doi.org/10.3390/s16020255, 2016.

Wehrhan, M.; Sommer, M.: A Parsimonious Approach to Estimate Soil Organic Carbon Applying Unmanned Aerial System (UAS) Multispectral Imagery and the Topographic Position Index in a Heterogeneous Soil Landscape, Remote Sens-Basel 13 (18), 3557, https://doi.org/10.3390/rs13183557, 2021.

Wessolek, G.; Asseng, S.: Trade-off between wheat yield and drainage under current and climate change conditions in northeast Germany, Eur. J. Agron. 24 (4), 333–342, https://doi.org/10.1016/j.eja.2005.11.001, 2006.

Wilken, F.; Ketterer, M.; Koszinski, S.; Sommer, M.; Fiener, P.: Understanding the role of water and tillage erosion from [239+240]Pu tracer measurements using inverse modelling, SOIL 6 (2), 549–564, https://doi.org/10.5194/soil-6-549-2020, 2020.

Xu, T.; Guo, Z.; Liu, S.; He, X.; Meng, Y.; Xu, Z.; Xia, Y.; Xiao, J.; Zhang, Y.; Ma, Y.; Song, L.: Evaluating Different Machine Learning Methods for Upscaling Evapotranspiration from Flux Towers to the Regional Scale, J. Geophys. Res-Atmos. 123 (16), 8674–8690, https://doi.org/10.1029/2018JD028447, 2018.


**Tables:**

**Table 1:** Performance classes to evaluate modeling approaches.

| Class | MAE | NRMSE | NSE | R2 |
|---|---|---|---|---|
| Very Good | < 0.35 | < 30 | > 0.85 | > 0.85 |
| Good | 0.35 <= 0.67 | 30 <= 40 | 0.85 => 0.75 | 0.85 => 0.75 |
| Satisfactory | 0.67 <= 1 | 40 <= 50 | 0.75 => 0.5 | 0.75 => 0.6 |
| Not Satisfactory | > 1 | > 50 | < 0.5 | < 0.6 |


**Table 2:** Calibration statists of all modeling approaches and treatments.

| Approach | MAE | NRMSE | NSE | R2 | Approach | MAE | NRMSE | NSE | R2 |
|---|---|---|---|---|---|---|---|---|---|
| **LL-cv n-d** | | | | | **LL-cv d** | | | | |
| MDV | 0.27 | 39.6 | 0.84 | 0.85 | MDV | 0.25 | 35.2 | 0.88 | 0.88 |
| LUT | 0.09 | 14.3 | 0.98 | 0.98 | LUT | 0.08 | 13.4 | 0.98 | 0.98 |
| NLR | 0.56 | 49.0 | 0.76 | 0.77 | NLR | 0.53 | 46.8 | 0.78 | 0.79 |
| SVM | 0.26 | 25.7 | 0.93 | 0.93 | SVM | 0.23 | 23.0 | 0.95 | 0.95 |
| ANN_BR | 0.31 | 28.0 | 0.92 | 0.92 | ANN_BR | 0.27 | 25.7 | 0.93 | 0.93 |
| **LL-ng n-d** | | | | | **LL-ng d** | | | | |
| MDV | 0.25 | 29.3 | 0.91 | 0.92 | MDV | 0.25 | 30.6 | 0.91 | 0.91 |
| LUT | 0.09 | 13.6 | 0.98 | 0.98 | LUT | 0.10 | 14.6 | 0.98 | 0.98 |
| NLR | 0.54 | 41.1 | 0.83 | 0.84 | NLR | 0.55 | 40.6 | 0.84 | 0.84 |
| SVM | 0.28 | 22.9 | 0.95 | 0.95 | SVM | 0.29 | 23.6 | 0.94 | 0.94 |
| ANN_BR | 0.31 | 24.6 | 0.94 | 0.94 | ANN_BR | 0.32 | 25.0 | 0.94 | 0.94 |
| **RG-ca n-d** | | | | | **RG-ca d** | | | | |
| MDV | 0.26 | 30.9 | 0.90 | 0.91 | MDV | 0.22 | 29.8 | 0.91 | 0.91 |
| LUT | 0.09 | 15.7 | 0.98 | 0.98 | LUT | 0.09 | 14.3 | 0.98 | 0.98 |
| NLR | 0.50 | 42.4 | 0.82 | 0.83 | NLR | 0.48 | 41.8 | 0.82 | 0.83 |
| SVM | 0.26 | 25.6 | 0.93 | 0.93 | SVM | 0.23 | 23.4 | 0.95 | 0.95 |
| ANN_BR | 0.30 | 27.3 | 0.93 | 0.93 | ANN_BR | 0.29 | 26.0 | 0.93 | 0.93 |


**Table 3:** Validation statists of all modeling approaches and treatments.

| Approach | MAE | NRMSE | NSE | R2 | Approach | MAE | NRMSE | NSE | R2 |
|---|---|---|---|---|---|---|---|---|---|
| **LL-cv n-d** | | | | | **LL-cv d** | | | | |
| **MDV** | 0.33 | 46.0 | 0.79 | 0.81 | **MDV** | 0.26 | 35.8 | 0.87 | 0.88 |
| **LUT** | 0.74 | 69.8 | 0.51 | 0.51 | **LUT** | 0.72 | 70.0 | 0.51 | 0.51 |
| **NLR** | 0.57 | 50.9 | 0.74 | 0.75 | **NLR** | 0.55 | 48.8 | 0.76 | 0.77 |
| **SVM** | 0.34 | 33.7 | 0.89 | 0.89 | **SVM** | 0.31 | 32.1 | 0.90 | 0.90 |
| **ANN_BR** | 0.35 | 32.2 | 0.90 | 0.90 | **ANN_BR** | 0.32 | 29.6 | 0.91 | 0.91 |
| **LL-ng n-d** | | | | | **LL-ng d** | | | | |
| **MDV** | 0.31 | 32.1 | 0.90 | 0.90 | **MDV** | 0.31 | 33.3 | 0.89 | 0.89 |
| **LUT** | 0.78 | 62.7 | 0.61 | 0.61 | **LUT** | 0.83 | 64.7 | 0.58 | 0.58 |
| **NLR** | 0.54 | 41.7 | 0.83 | 0.83 | **NLR** | 0.55 | 41.5 | 0.83 | 0.84 |
| **SVM** | 0.32 | 25.4 | 0.94 | 0.94 | **SVM** | 0.33 | 26.5 | 0.93 | 0.93 |
| **ANN_BR** | 0.33 | 25.9 | 0.93 | 0.93 | **ANN_BR** | 0.34 | 26.6 | 0.93 | 0.93 |
| **RG-ca n-d** | | | | | **RG-ca d** | | | | |
| **MDV** | 0.28 | 34.4 | 0.88 | 0.89 | **MDV** | 0.27 | 31.8 | 0.90 | 0.90 |
| **LUT** | 0.79 | 71.8 | 0.48 | 0.49 | **LUT** | 0.69 | 65.9 | 0.57 | 0.57 |
| **NLR** | 0.49 | 42.2 | 0.82 | 0.83 | **NLR** | 0.48 | 42.0 | 0.82 | 0.83 |
| **SVM** | 0.29 | 28.1 | 0.92 | 0.92 | **SVM** | 0.26 | 26.2 | 0.93 | 0.93 |
| **ANN_BR** | 0.33 | 29.9 | 0.91 | 0.91 | **ANN_BR** | 0.31 | 28.8 | 0.92 | 0.92 |




 **Figures:**

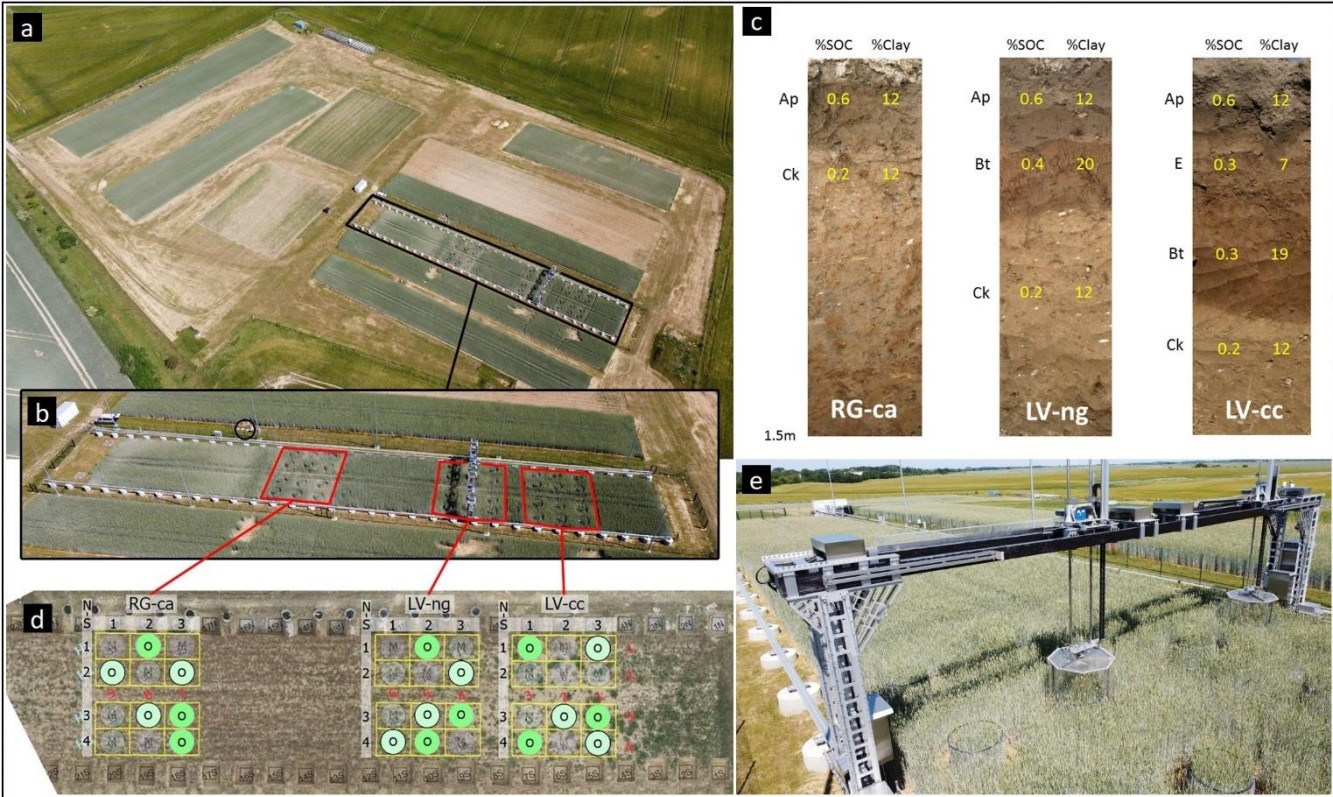

**Figure 1:** (a) AgroFLUX research site in the CarboZALF-D experimental area with (b) the 110 x 16 m field where (c) three different soil types (LV-cc: non-eroded calcic Luvisol; LV-ng: highly eroded nudiargic Luvisol; and RG-ca: extremely eroded calcaric Regosol) (c) 18 measurement plots of (e) the FluxCrane operates on. Soil moisture and precipitation measurements were taken in the marked area (black circle, b). The separation of non-diluted (unframed green) and diluted (framed light green) plots can be seen in (c).

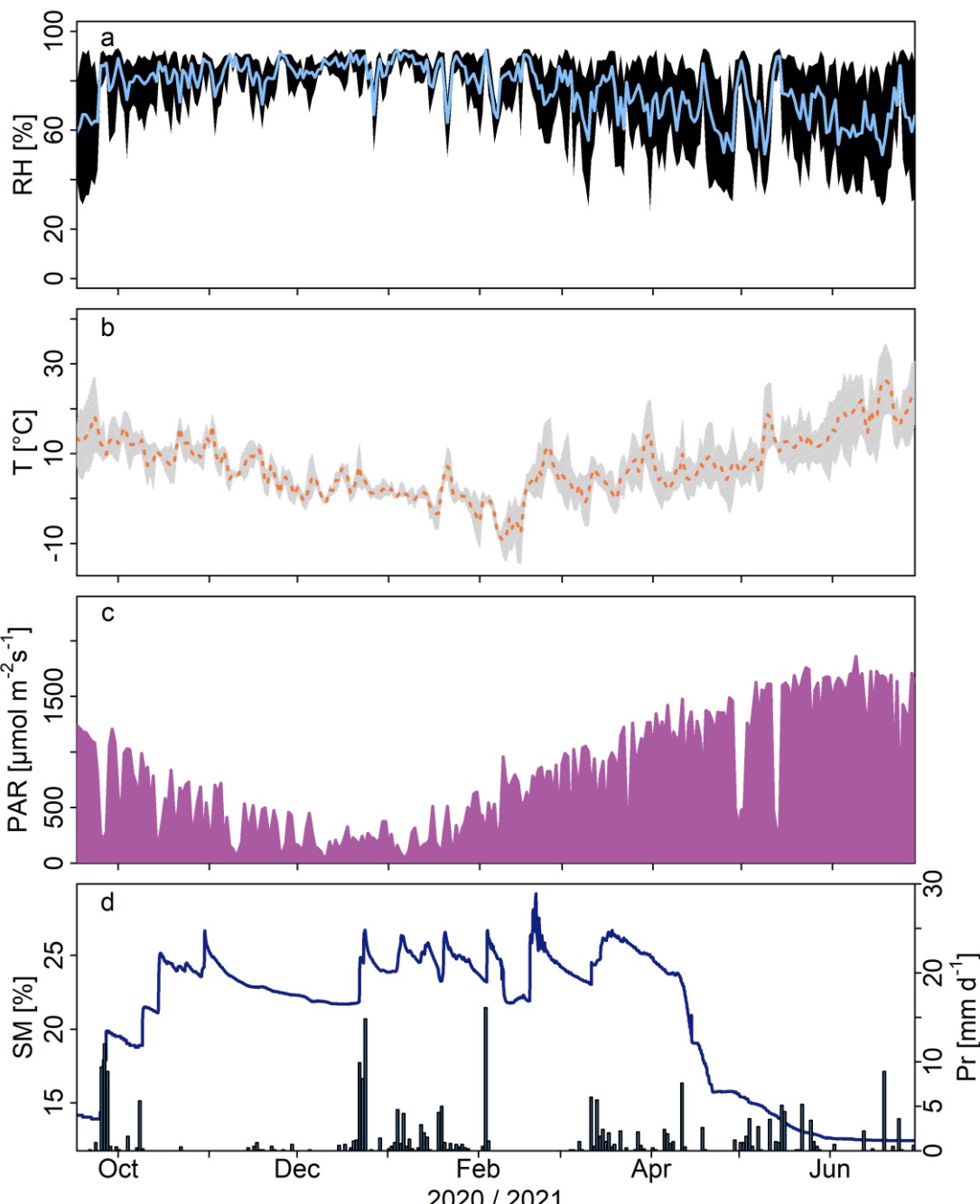


**Figure 2:** Environmental parameters during the measurement period with (a) daily mean relative humidity (RH; black line; dotted lines = corresponding variation), (b) daily mean temperatures (T; orange line; light gray = corresponding variation), (c) incoming photosynthetically active radiation

(PAR; purple) and (d) soil moisture (SM; blue line) and precipitation (PR; blue bars).

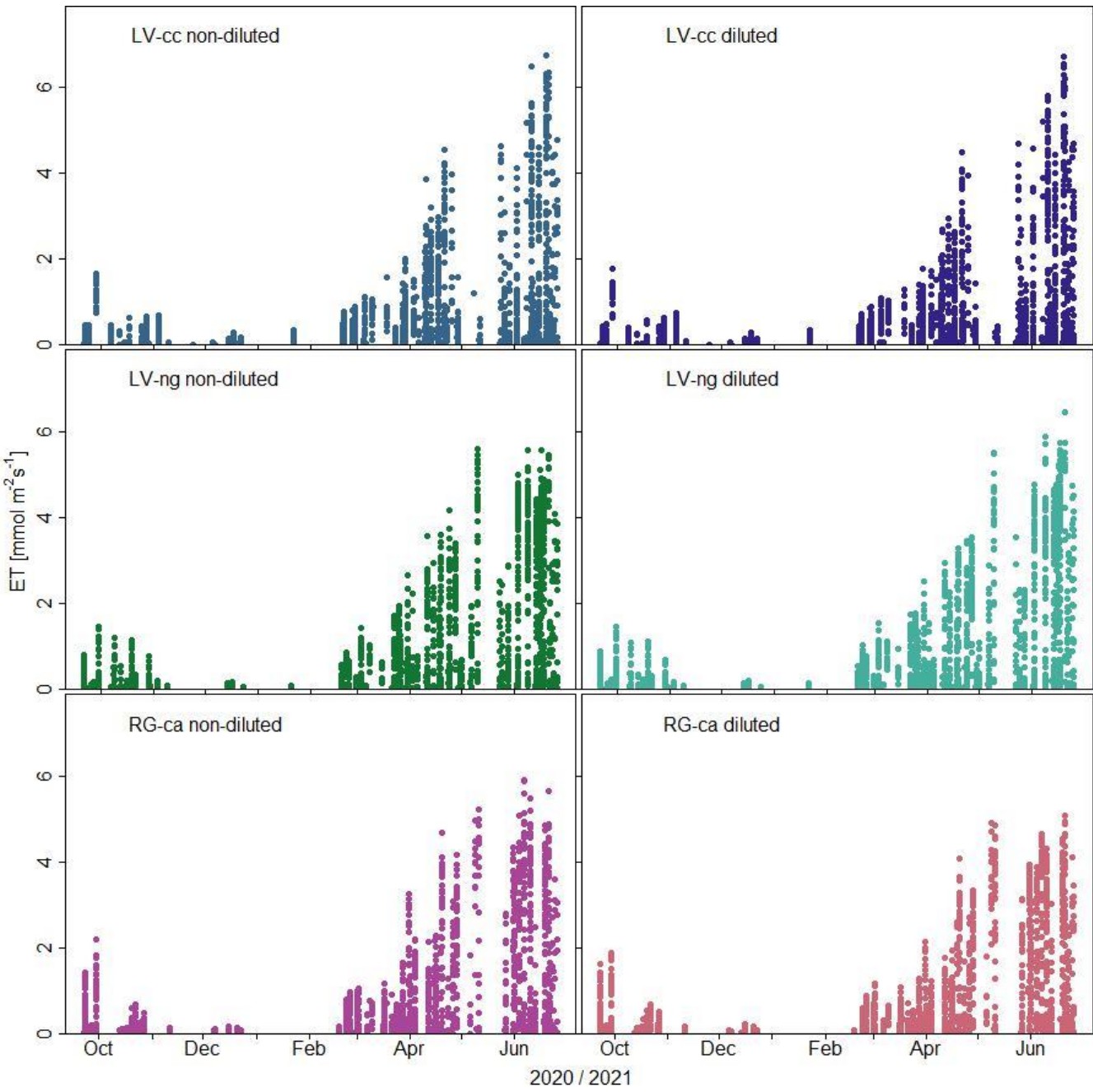

**Figure 3:** Measured and quality-screened (by soft and hard criteria) ET fluxes of the three soil types over the entire observation period (non-diluted treatments on the left, diluted treatments on the right).

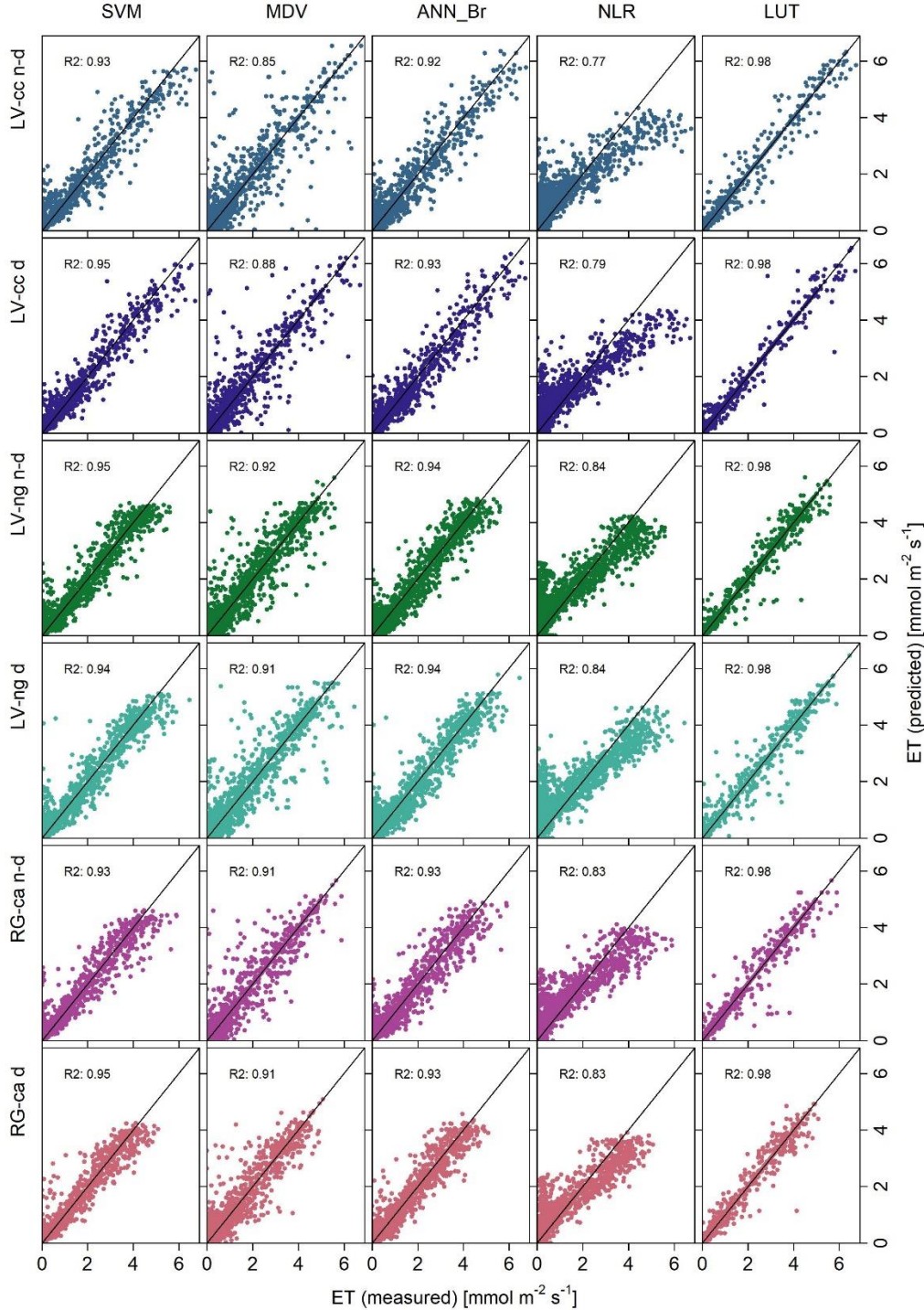

**Figure 4:** Comparison of the measured and predicted ET fluxes and associated r-squared values (R2) of the calibration results of all modeling approaches. The black line represents the 1/1 line. The different modeling approaches are shown on top, the treatments on the left.

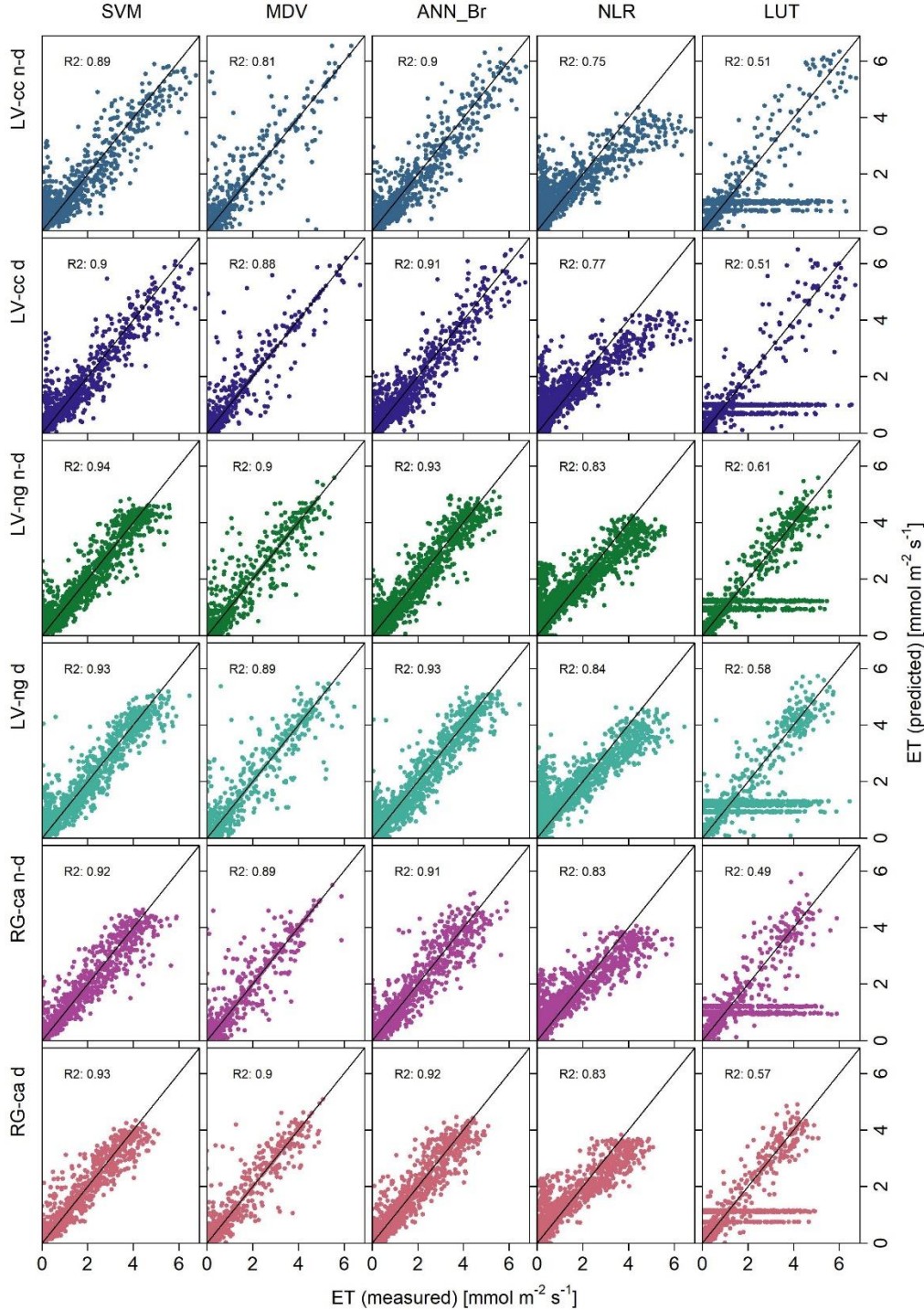

**Figure 5:** Comparison of the measured with the predicted ET fluxes and the associated r-squared values (R2) of the validation results of all modeling approaches. The black line represents the 1/1 line. The different modeling approaches are shown on top, the treatments on the left.

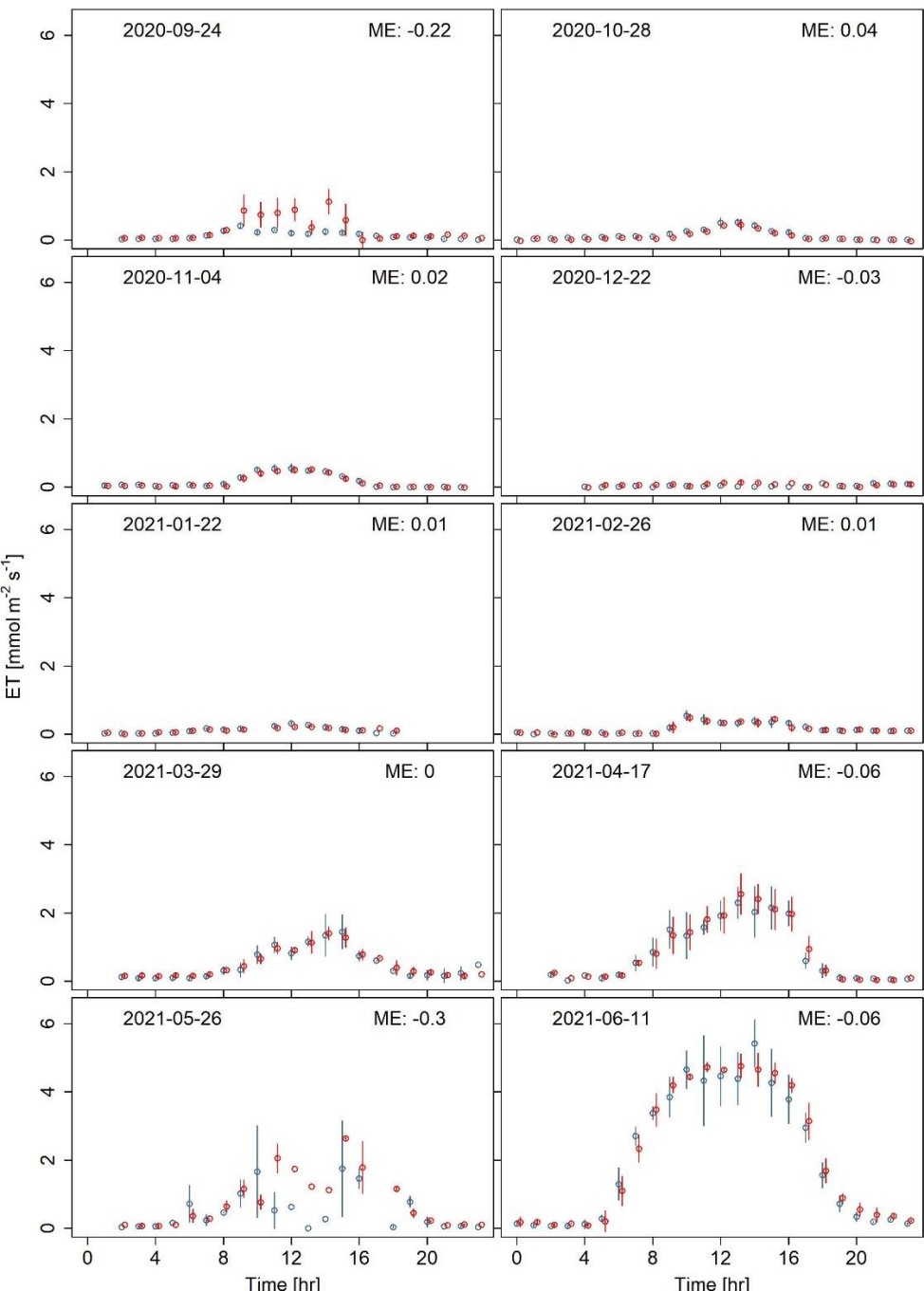

**Figure 6**: Diurnal cycles in ET fluxes during the cultivation season for one sample day per month (day with the most measurements) and corresponding mean error (ME; two digits rounded). Measured ET is shown in blue, predicted ET is shown in red.

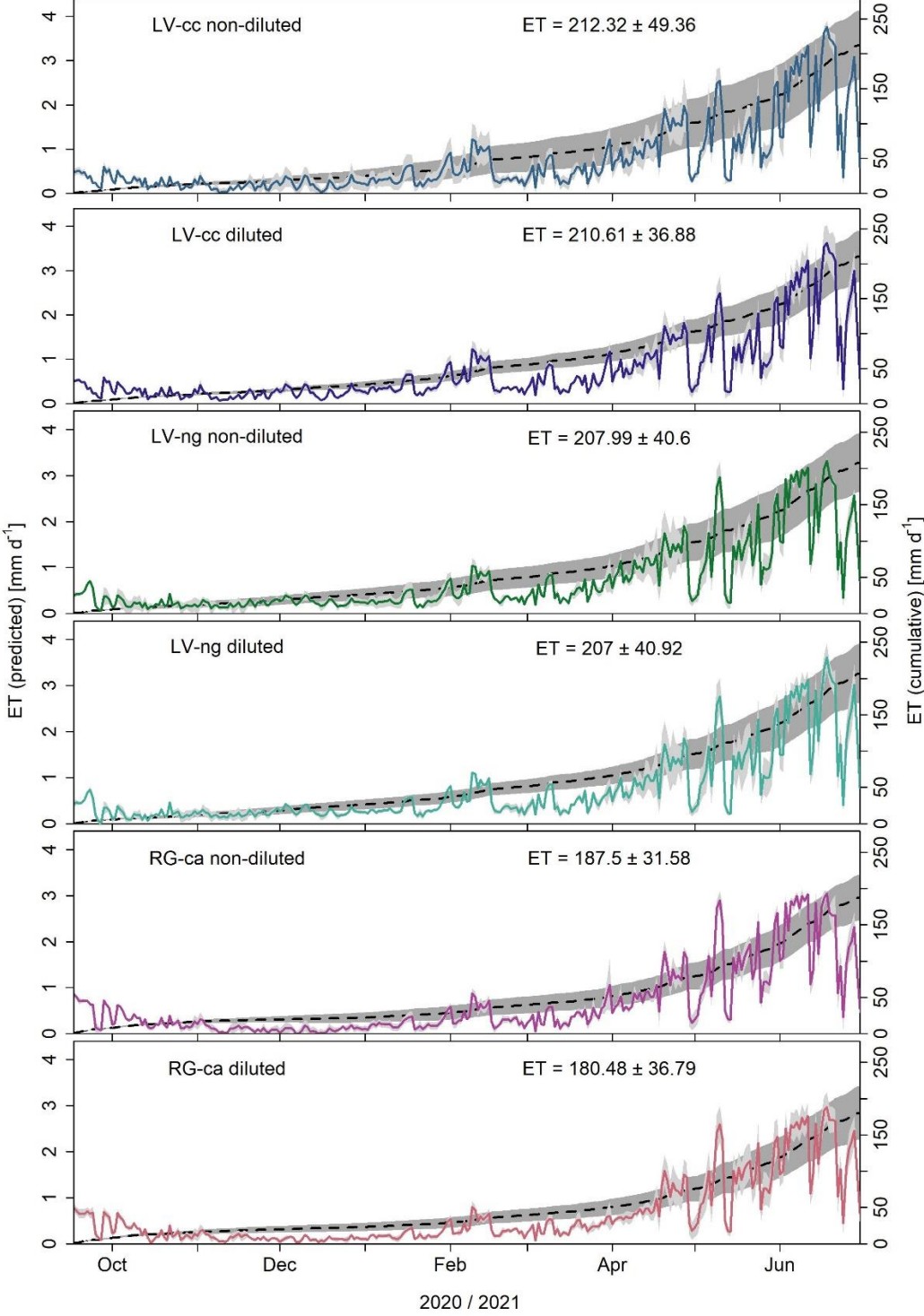


**Figure 7:** Daily mean ET sums (colored lines) of the different treatments and seasonal cumulative ET (ET$_{sum}$; dashed lines) with standard deviation between replicates (light and dark gray).

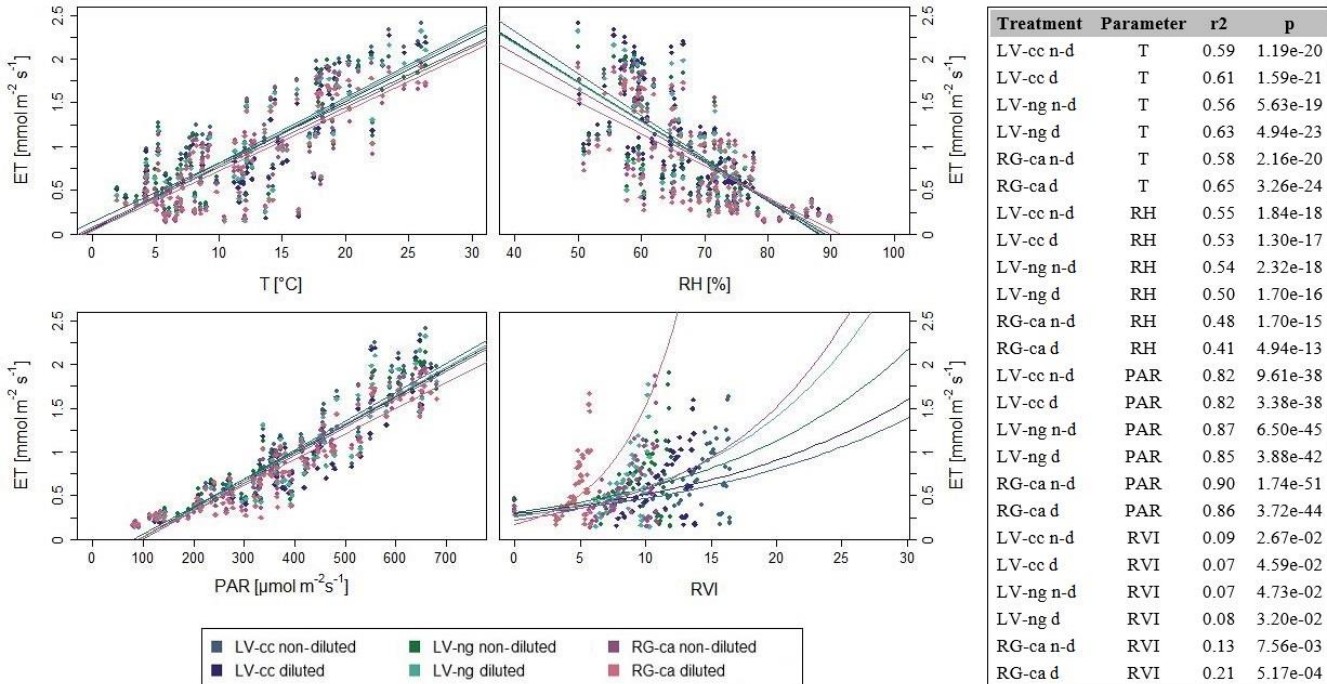

| Treatment | Parameter | r2 | p |
|-----------|-----------|------|---------|
| LV-cc n-d | T | 0.59 | 1.19e-20 |
| LV-cc d | T | 0.61 | 1.59e-21 |
| LV-ng n-d | T | 0.56 | 5.63e-19 |
| LV-ng d | T | 0.63 | 4.94e-23 |
| RG-ca n-d | T | 0.58 | 2.16e-20 |
| RG-ca d | T | 0.65 | 3.26e-24 |
| LV-cc n-d | RH | 0.55 | 1.84e-18 |
| LV-cc d | RH | 0.53 | 1.30e-17 |
| LV-ng n-d | RH | 0.54 | 2.32e-18 |
| LV-ng d | RH | 0.50 | 1.70e-16 |
| RG-ca n-d | RH | 0.48 | 1.70e-15 |
| RG-ca d | RH | 0.41 | 4.94e-13 |
| LV-cc n-d | PAR | 0.82 | 9.61e-38 |
| LV-cc d | PAR | 0.82 | 3.38e-38 |
| LV-ng n-d | PAR | 0.87 | 6.50e-45 |
| LV-ng d | PAR | 0.85 | 3.88e-42 |
| RG-ca n-d | PAR | 0.90 | 1.74e-51 |
| RG-ca d | PAR | 0.86 | 3.72e-44 |
| LV-cc n-d | RVI | 0.09 | 2.67e-02 |
| LV-cc d | RVI | 0.07 | 4.59e-02 |
| LV-ng n-d | RVI | 0.07 | 4.73e-02 |
| LV-ng d | RVI | 0.08 | 3.20e-02 |
| RG-ca n-d | RVI | 0.13 | 7.56e-03 |
| RG-ca d | RVI | 0.21 | 5.17e-04 |

**Figure 8:** Relationship between ET and temperature (T) [°C], relative humidity (RH) [%], photosynthetically active radiation (PAR) [μmol $m^{-2}$ $s^{-1}$], and ratio vegetation index (RVI) [mmol $m^{-2}$ $s^{-1}$], and associated regression lines. Statistical values (r² and p) for the relationship between ET and response variables (environmental parameters) are presented in the table.



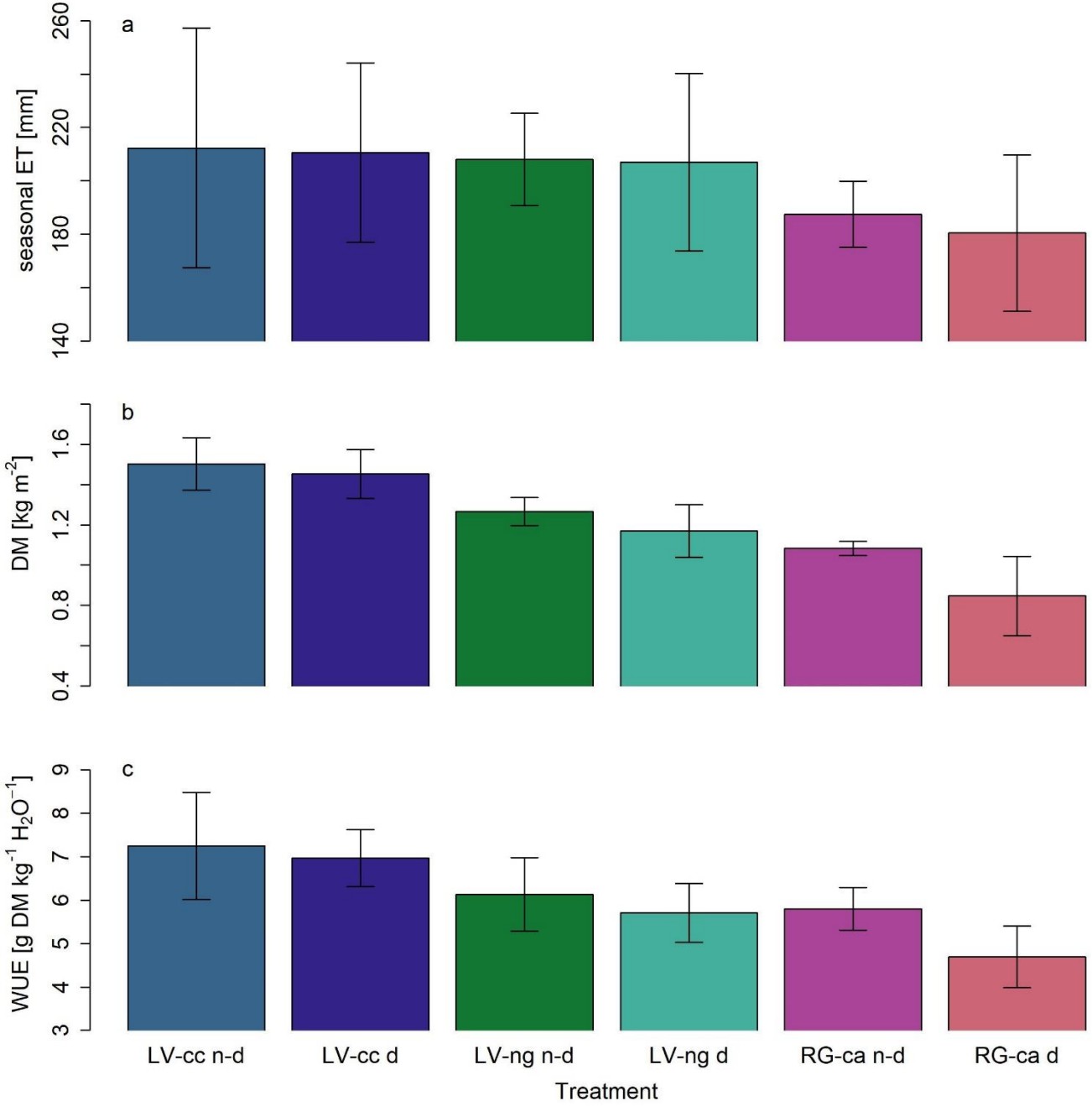

**Figure 9:** Averaged seasonal cumulative ET (ET$_{sum}$) [mm] (a), harvest in form of dry mass (DM) [kg] (b), and WUE$_{agro}$ of the different treatments and the associated standard deviation.



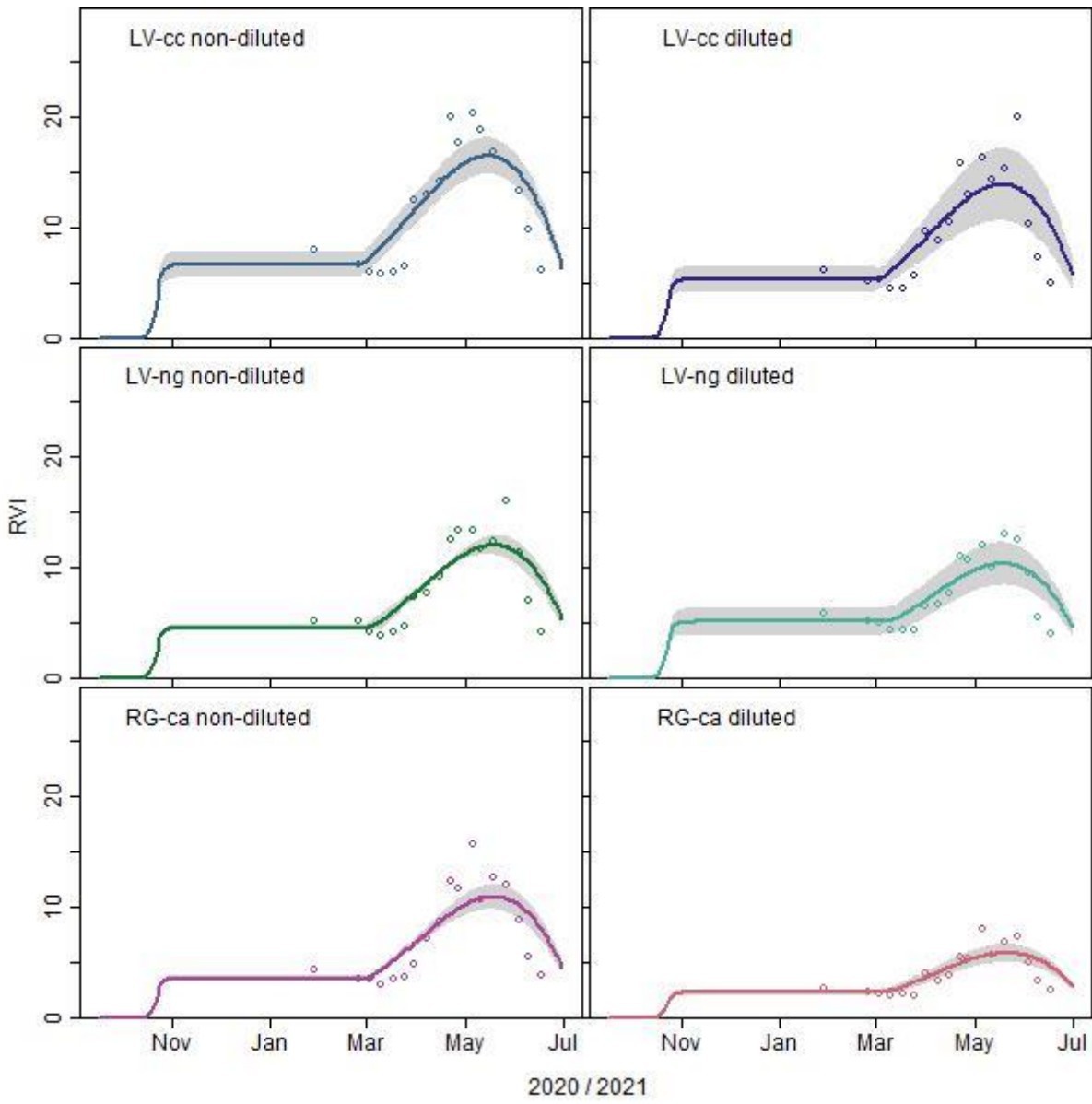

**Figure A1:** RVI fit (colored lines) of the different treatments with the standard deviation between replicates (light gray) and the corresponding averages of the daily measurements (points).


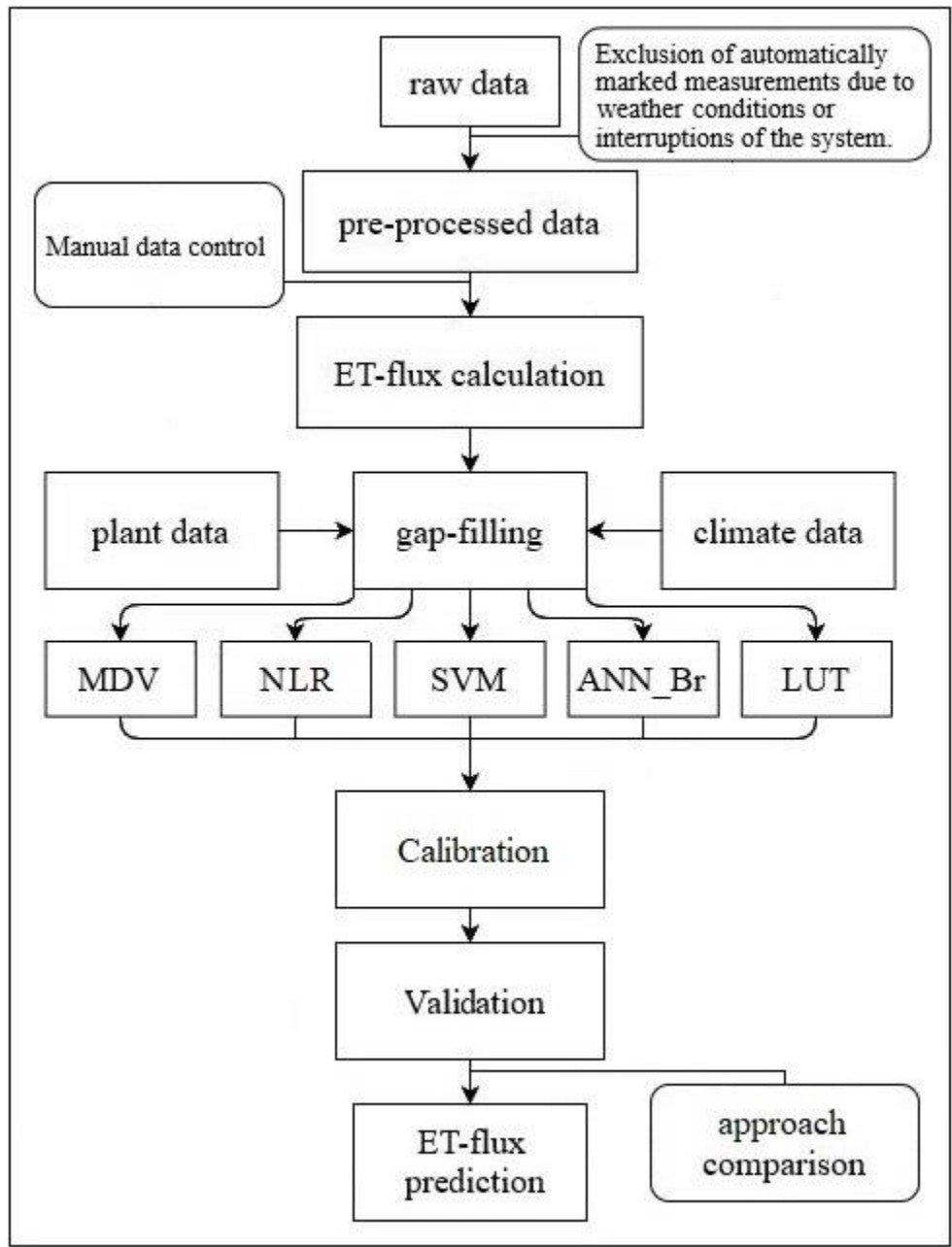

**Figure A2:** Schematic representation of the main steps of the presented data processing: raw data preparation was followed by a campaign-specific ET-flux calculation. Then, environmental parameters were used for modeling using five different approaches. After calibration and validation, the most accurate approach was used for ET flux modeling.


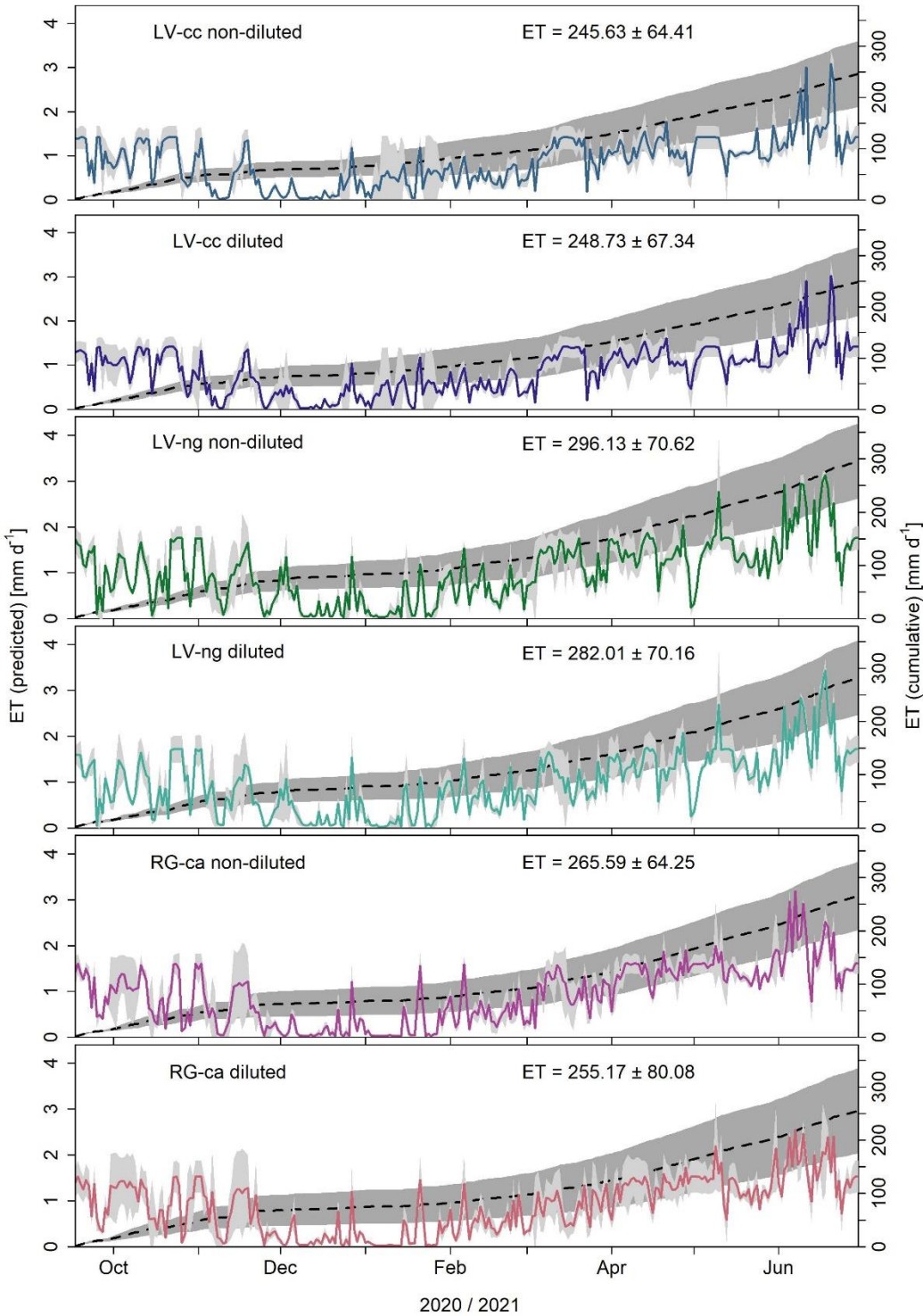

**Figure A3:** LUT predicted daily mean ET sums (colored lines) of the different treatments and seasonal cumulative ET (ET$_{sum}$; dashed lines) with standard deviation between replicates (light and dark gray).

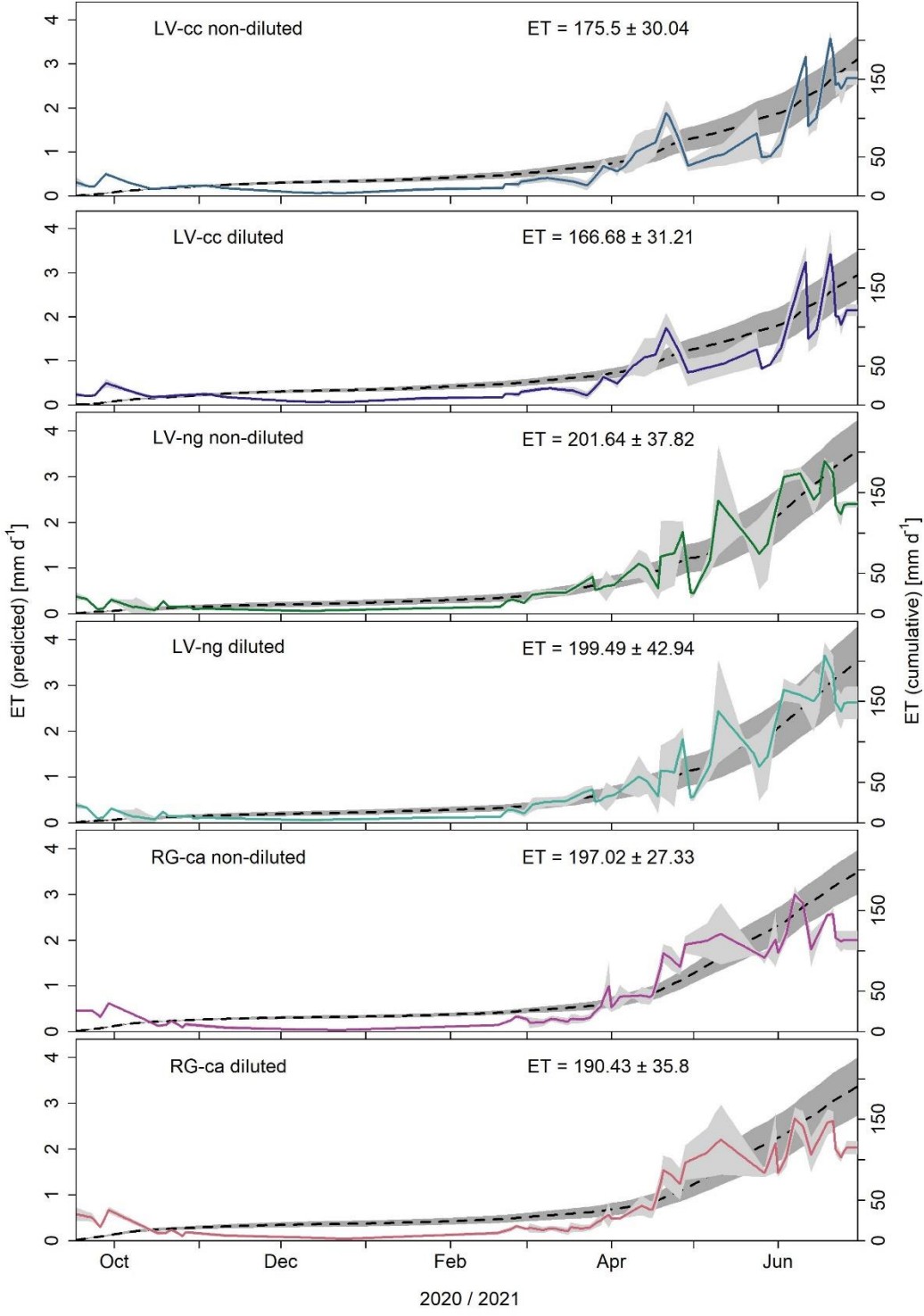

**Figure A4:** MDV predicted daily mean ET sums (colored lines) of the different treatments and seasonal cumulative ET (ET$_{sum}$; dashed lines) with standard deviation between replicates (light and dark gray).

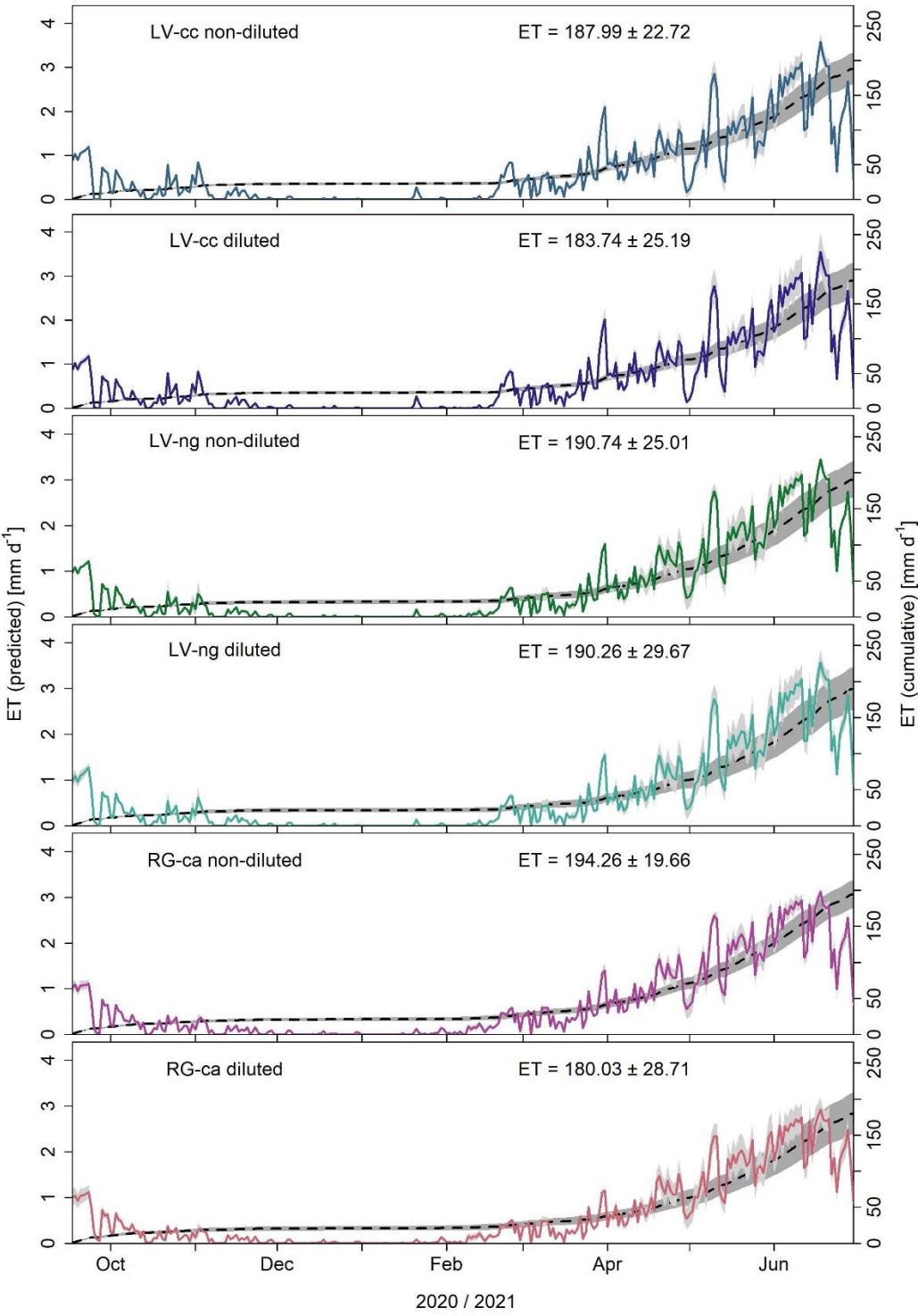


**Figure A5:** NLR predicted daily mean ET sums (colored lines) of the different treatments and seasonal cumulative ET (ET$_{sum}$; dashed lines) with standard deviation between replicates (light and dark gray).

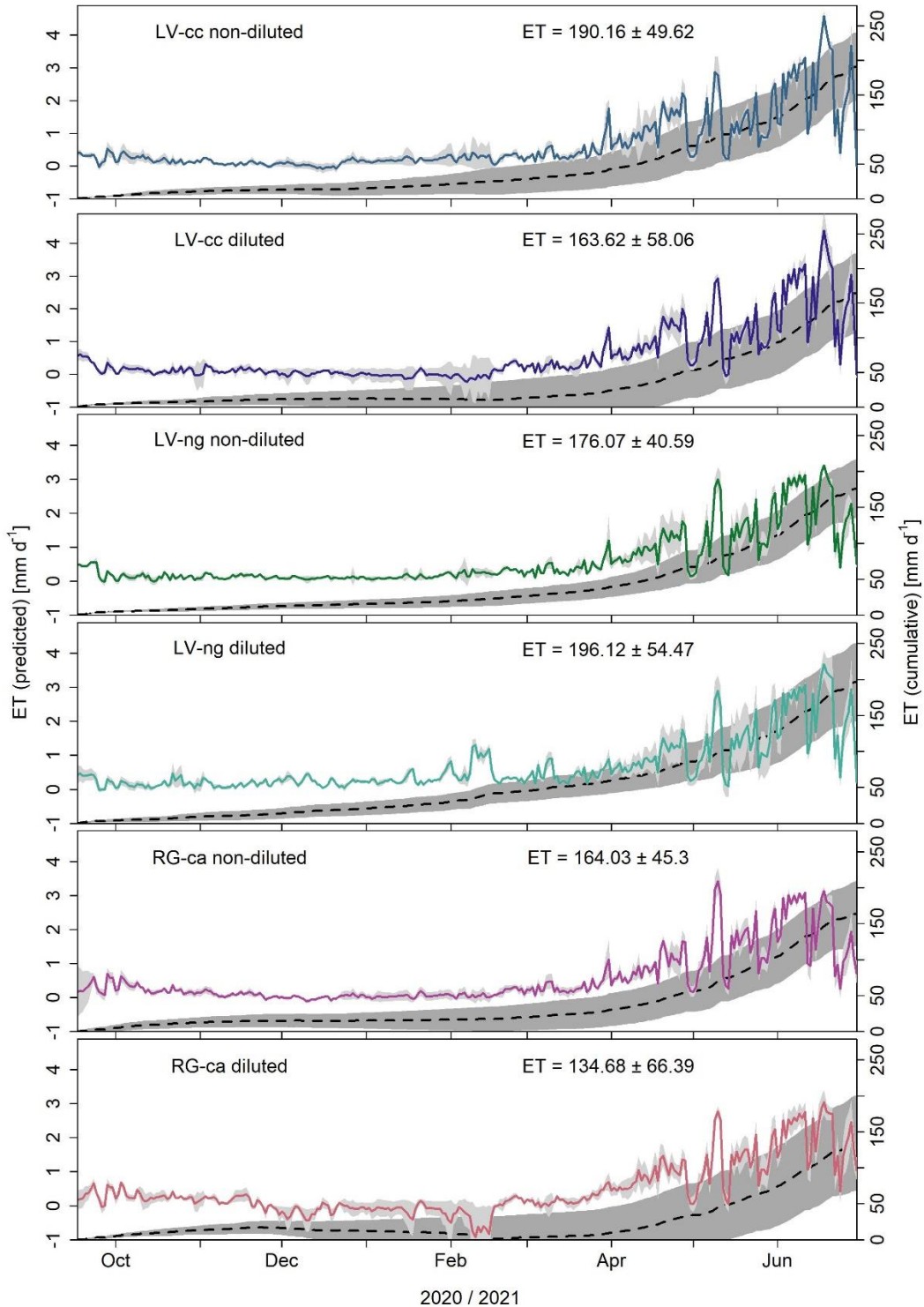

**Figure A6:** ANN_BR predicted daily mean ET sums (colored lines) of the different treatments and seasonal cumulative ET (ET$_{sum}$; dashed lines) with standard deviation between replicates (light and dark gray).


**Table B1:** Fertilization information for the field.

| Date | Amount | Details |
|------|--------|---------|
| 15.10.2020 | 161 kg $P_2O_5$ ha$^{-1}$ | applied on 6 plots of LL as TSP |
| 22.03.2020 | 77  kg $P_2O_5$ ha$^{-1}$ | as Triple Super Phosphate (TSP) |
| 22.03.2020 | 259 kg $K_2O$ ha$^{-1}$ | as 40% grain potash |
| 16.09.2020 | 30  kg N ha$^{-1}$ | 10 m³ ha$^{-1}$ digestate |
| 10.03.2021 | 91  kg N ha$^{-1}$ | 30 m³ ha$^{-1}$ digestate |
| 08.04.2021 | 45  kg N ha$^{-1}$ | 12 m³ ha$^{-1}$ digestate |


**Table B2:** The number of measurements per treatment and the percentage of modeling data.

| Plot | | Measurements [n] | modelled [%] |
|------|---|------------------|--------------|
| LV-cc n-d | 1 | 990 | 85.63 |
| LV-cc n-d | 2 | 624 | 90.94 |
| LV-cc n-d | 3 | 996 | 85.54 |
| LV-cc d | 1 | 624 | 90.94 |
| LV-cc d | 2 | 735 | 89.33 |
| LV-cc d | 3 | 989 | 85.64 |
| LV-ng n-d | 1 | 1210 | 82.43 |
| LV-ng n-d | 2 | 1210 | 82.43 |
| LV-ng n-d | 3 | 705 | 89.76 |
| LV-ng d | 1 | 718 | 89.58 |
| LV-ng d | 2 | 1215 | 82.36 |
| LV-ng d | 3 | 1205 | 82.51 |
| RG-ca n-d | 1 | 657 | 90.46 |
| RG-ca n-d | 2 | 772 | 88.79 |
| RG-ca n-d | 3 | 669 | 90.29 |
| RG-ca d | 1 | 669 | 90.29 |
| RG-ca d | 2 | 1130 | 83.59 |
| RG-ca d | 3 | 1129 | 83.61 |


**Table B3:** Used R packages and associated sources.

| package | source |
|---|---|
| Akima | Akima & Gebhardt (2021) |
| Andrews | Myslivec (2012) |
| Base | R Core Team (2021) |
| Boot | Davison & Hinkley (1997) |
| Caret | Kuhn (2021) |
| data.table | Dowle & Srinivasan (2021) |
| e1071 | Meyer et al. (2021) |
| FSA | Ogle et al. (2022) |
| ggplot2 | Wickham (2016) |
| gridExtra | Auguie (2017) |
| gt | Iannone et al. (2022) |
| hydroGOF | Mauricio Zambrano-Bigiarini (2020) |
| Kernlab | Karatzoglou et al. (2004) |
| Lattrice | Sarkar (2008) |
| Lmtest | Zeileis & Hothorn (2002) |
| lookupTable | Jia & Maier (2015) |
| Lubridate | Grolemund & Wickham (2011) |
| Neuralnet | Fritsch et al. (2019) |
| Nortest | Gross & Ligges (2015) |
| Plotrix | J (2006) |
| Plyr | Wickham (2011) |
| Reshape | Wickham (2007) |
| Shape | Soetaert (2021) |
| Tibble | Müller & Wickham (2021) |
| tidyr | Wickham & Girlich (2022) |
| Vioplot | Adler & Kelly (2020) |
| webshot | Chang (2022) |
| Zoo | Zeileis & Grothendieck (2005) |

**Table B4:** Calibration mean error (ME) for different ranges of ET fluxes (less than 2, between 2 and 4 and greater than 4 mmol m$^{-2}$ s$^{-1}$) for all modeling approaches.

| Approach | < 2 | 2 - 4 | > 4 |
|---|---|---|---|
| SVM | -0.05 | 0.05 | 0.4 |
| MDV | -0.03 | 0.06 | 0.29 |
| ANN_BR | -0.03 | 0.08 | 0.33 |
| NLR | -0.2 | 0.39 | 1.14 |
| LUT | -0.01 | 0.01 | 0.12 |

975