# Peer review of "Benefits of a robotic chamber system for determining evapotranspiration in an erosion affected, heterogeneous cropland"

_Hydrology and Earth System Sciences, 2022_

## Author Response (AR1)

To the Editor:

Attached please find our revised manuscript now entitled "*Benefits of a robotic chamber system for determining evapotranspiration in an erosion affected, heterogeneous cropland*" to be considered for publication.

This work has been previously submitted to HESS, with "major revisions" being suggested to apply to our manuscript. The main recommendations of both reviewers included: spell checking the manuscript and subject it to a native speaker correction; reworking the aims/hypotheses and discussion section; shortening and sharpening the conclusions; expanding the methods section.

We have significantly revised our manuscript in accordance with these suggestions, taking all criticism and comments into account. Particularly, we emphasize the dual focus of our work more clearly. We reformulated the aims and introduced hypotheses that reflect our focus on both the methodological (novel FluxCrane system, gap filling approaches) and ecohydrological (impact of soil heterogeneity and management on ET and ET flux dynamics). We have largely reworked the entire manuscript and particularly the discussion to reflect our aims and hypotheses as suggested by both reviewers. Moreover, we have reworked the conclusions section, which now only summarizes our main findings. In addition, we would like to note that the revision process alerted us to a minor error in our flux calculations, which we have corrected for the revised version. This led to a small systematic increase in $ET_{sum}$ of all plots (by ~10 mm), which however doesn't change any statements or messages of our manuscript.

In conclusion, we believe the implemented revisions improved the manuscript significantly and thank you very much for your consideration.

Sincerely,

Adrian Dahlmann

Review of "Measuring evapotranspiration on an eroded cropland by an automated and mobile chamber system: gap filling strategies and impact of soil type and topsoil removal" by Adrian Dahlmann et al., for HESS.

Summary

This manuscript presents a study of roughly one year of evapotranspiration measurements of a winter rye crop on a number of soil types. The ET measurements are novel, using an automated and mobile chamber system. The study reports and highlights the negative impact of eroded conditions on biomass growth and water use efficiency. Because of the novelty of the ET method, the study also investigates different gap-filling techniques. Here they find that machine learning approaches are better than simpler regression or look-up techniques. While this is an interesting study, I think there are many improvements that can be made to give it more relevance to both application users (farmers, land managers) and to the scientific community.

*Thank you for your time and effort in reviewing our manuscript. We appreciate your constructive comments and have taken all of them into consideration. A detailed reply is posted below each comment in italics for better visibility.*

**Major comments**

1. The focus of the paper is not clear and wanders between the new ET method and its gap-filling, the erosion gradient of soils, the role of ET in the water balance, and other impacts of erosion (e.g., on biomass production). A tighter, clearer focus will help – and then the methods parts can better be written to support the key aim, which is (to me at least) to explore the impact of soil conditions on agronomic and water balance impacts.

2. Hypothesis-testing should be performed – it should be clearer both in the abstract and end of the introduction what the expectation is – how should the different soils change agronomic and ET outcomes? How should we be able to anticipate, from the literature, different performance among the gap-filling techniques?

*The aim of our manuscript was indeed threefold:*
   1. *To shed light on the impact of eroded soils and top-soil dilution on cumulative seasonal ET.*
   2. *The spatio-temporal variability of ET flux and WUEagro.*
   3. *To describe our new set up and test it with a particular focus on gap filling strategies.*

*We fully agree that in the previous version these three aims (and corresponding hypotheses) might not have been clearly phrased.*

*We regard all of these aspects as important. Hence, instead of cutting out one of them from the manuscript, we have rephrased the aims and added corresponding hypotheses to each aim in the Introduction, and carefully revised the entire manuscript to adhere strictly to the structure we define with our aims and hypotheses. We believe that this has yielded the needed clarity to follow each aspect of the manuscript.*

3. In general, there are many mistakes in the English language related to article use, comma positions, sentence structure, possessive apostrophe, etc. I have noted some of these below, but a careful edit should be performed with these issues in mind.

*Thank you for your suggestion. The revised manuscript was carefully read and spell-checked by a native speaking colleague, whose contribution we added to the acknowledgements.*

**Minor comments**

The intro could be condensed, particularly the first paragraph (e.g., L36-40, 47-52, and remove "according to the European Union").

*We have condensed the introduction. Particularly we have:*

- *Removed first sentence.*
- *Removed second part of second sentence.*
- *Removed "According to the European Union"*
- *Connected two sentences about soil degradation*

Consider moving L53-62 to the site description section in the methods.

*Done. It now reads: "Brandenburg, which includes some of the driest regions in Germany, uses 48.6% or about 1.44 million hectares of its area for agriculture (Amt für Statistik Berlin-Brandenburg, 2020). It is located in the continental climate zone and has a water deficit of about 150 mm during the growing season (Wessolek and Asseng 2006)."*

*and:*

*"The focus of agriculture in Brandenburg is on grain production, which faces a variety of challenges due to increasingly dry conditions during the main growing season (Amt für Statistik Berlin-Brandenburg, 2020)."*

L77-81, Remove/condense; consider "field" instead of plot for eddy covarnacie

*Done. It now reads: "At the field scale for example, eddy covariance systems provide high frequency estimates of ET of a homogeneous system while currently dominant manual chamber approaches are able to precisely capture multi treatment effects (<1m²) on ET at the plot scale, but lack the high frequency."*

L103-5 consider at minimum three aims (i.e., separate soil type and management from spatio-temporal variability). In general note other comments that there should be clearer hypotheses in this area about expectations related to soil/management type, seasonal development of ET and WUE, and the gap-filling methods

*We fully agree that in the previous version our aims (and corresponding hypotheses) might not have been clearly phrased. Consequently, in addition to rephrasing the aims and adding corresponding hypotheses to each aim, we revised the entire manuscript to stick strictly to the structure we define with our aims and hypotheses.*

*It now reads: "In the following we will examine i) soil type and top soil dilution effects on crop yield, $ET_{sum}$ and $WUE_{agro}$, ii) the spatio-temporal variability of ET fluxes over the growing season, and iii) the suitability of various gap-filling strategies. The paper's aim is to establish an approach that would provide reliable predictions of ET fluxes both in terms of $ET_{sum}$ as well as diurnal trends of ET fluxes. We hypothesize that: i) eroded soils and top-soil dilution lead to decreased ET controlled by weaker*

*plant growth, ii) WUE$_{agro}$ declines from least to most eroded soil type and with top soil dilution; iii) the automated, continuous FluxCrane measurements result in unique insights into small scale dynamics such as night time ET fluxes and ET fluxes during the non-growing season. Here, we hypothesize, that iv) the uncommonly (compared to manual chamber systems) large data set allows for a robust use of gap-filling strategies based on machine learning. We envisage that this will greatly improve ET$_{sum}$ and subsequently WUE$_{agro}$ based on automated closed chamber systems."*

L141 what kind of digestate?

*We include a table with fertilization information in the appendix and refer to it in this section.*

*It now reads: "Fertilization was applied to all plots per soil type before and during the growing season using digestate from Pflanzenbauhof GbR (Uckermark, Germany), Triple Super Phosphate (TSP) and grain potash (Table B1)."*

| Date | Amount | Details |
|---|---|---|
| 15.10.2020 | 161 kg P$_2$O$_5$ ha$^{-1}$ | applied on 6 plots of LL as TSP |
| 22.03.2020 | 77 kg P$_2$O$_5$ ha$^{-1}$ | as Triple Super Phosphate (TSP) |
| 22.03.2020 | 259 kg K$_2$O ha$^{-1}$ | as 40% grain potash |
| 16.09.2020 | 30 kg N ha$^{-1}$ | 10 m³ ha$^{-1}$ digestate |
| 10.03.2021 | 91 kg N ha$^{-1}$ | 30 m³ ha$^{-1}$ digestate |
| 08.04.2021 | 45 kg N ha$^{-1}$ | 12 m³ ha$^{-1}$ digestate |

Section 2.3 misses info on measurement times, accuracy, precision, etc. of this novel system

*We agree that some information is missing or mentioned in the wrong place. All information concerning the system (mainly from chapter "2.5.1 ET Flux calculation") and other missing information is now incorporated.*

L157 describe the adjacent field a bit – given that the soils and management status can drastically change the soil moisture, how was the adjacent field treated? Also please add where this field is to the site map.

*Sorry for the misleading description, it was not an adjacent field it was a plot on the same field. We have now marked the adjacent plot in Fig. 1 (< 25m away). We added the location to the site map. It now reads: "Precipitation (Pr) [mm] (Tipping Bucket Rain Gauge 52203, R. M. Young Company, USA) and soil moisture (SM; 13 to 18 cm depth) [%] (ML2x, Delta-T Devices Ltd., UK) were measured at an adjacent field (< 25m; Fig. 1)."*

L180 some chamber info is missing – what is the headspace, how tall is it relative to the biomass)

*See above comment on section 2.3.*

*The maximum plant height was 1.35m to 1.65m depending on the treatment. At a chamber height of 1.94m there was a minimum headspace of 29 cm.*

L177 consider moving all or some of this on flux calculations and gap-filling to section 2.3

*Done. It's now part of 2.5.1.*

L214 clarify if there is a moving window to this NLR or it's just clumping all the data – if so – how about trying a moving window or adding a term like days after planting (though RVI may be sufficient, it's not clear – were regressions of the residuals tested?).

*Indeed initially a variable moving window approach (joining 2 to up to 21 consecutive measurement days) was implemented for the NLR. However, due to limited amount of data, this approach performed less well than the NLR for the entire crop season (including RVI as parameter to account for crop growth).*

*It now reads: "NLR is based on parameterized non-linear equations using the mean least square method to express the relation between the total seasonal data of ET and T, RH, SM, PAR and RVI."*

L295 "differ between" and "minor differences" – are statistics performed here? Perhaps also some ranges of values can be given in the text

*This sentence was misleading: we were trying to describe the general trend of the measured fluxes. This section will be rewritten in the revised version also adding some value ranges.*

*It now reads: "The seasonal development (Fig. 3) of the quality-screened measured ET-fluxes is similar for all treatments: a short growth phase after germination (1 - 2 mmol m$^{-2}$ s$^{-1}$) is followed by a decrease of fluxes until and during the non-growing season in winter (< 0.1 mmol m$^{-2}$ s$^{-1}$), when hardly any plant activity is found due to low temperatures."*

L297 it's not clear what magnitude is being described

*We rephrased for better understanding. It now reads: "After the non-growing season, fluxes quickly return to their maximum fall level (1 - 2 mmol m$^{-2}$ s$^{-1}$) and then increase rapidly (> 5 mmol m$^{-2}$ s$^{-1}$)."*

L302 start with the result as a topic sentence, rather than the exposition. The result and not "figure 4" should lead the paragraph.

*We changed the formulation. It now reads: "Pronounced differences of tested gap-filling approaches in terms of respective calibration statistics could be found. Calibration model-performance differ in their scatter around the 1:1-agreement plots (Fig. 4) and associated coefficients of determination (R2)."*

L308 the allocation problems could be described / tested? (also "one can quickly see…" can be edited)

*We added a table of different performance classes in the methods section (2.7), including ranges for the statistical parameters that we used to define the quality of gap-filling approaches. We also changed the wording.*

*It now reads: "Considering the k-fold cross-validation (Fig. 5, Table 3), ANN_BR and SVM perform good, while MDV shows partially satisfactory statistics, and LUT shows unsatisfactory statistics due to allocation problems that arise when no class is found for the given climate conditions and the mean is used."*

L310 "a large number of negative ET fluxes" – first, how many? Second, is there evidence of dew? A negative ET isn't so implausible

*It's true that negative ET could be possible. Negative ET occurred only for gap-filled/modelled fluxes using ANN-Br in large amounts (35.5% for an example plot).This amount of negative fluxes, coupled with a magnitude of up to -6 mm/d (resulting in up to -51 mm of seasonal cumulative ET), cannot be explained by dew or uptake via leaves. In addition this occurred in an unlikely magnitude during periods where the air was not near saturation and the plants were active (e.g. during daylight). It is much more likely that this magnitude is an extrapolation problem of ANN-Br. We clarify this in the revised version.*

*It now reads: "However, gap-filled ET fluxes using ANN_BR showed a large number of predicted negative ET fluxes (1547 on average per plot; Figure 4, Supplement) throughout the cultivation period. These fluxes occurred to an unrealistic degree during times when RH was significantly below saturation and plants were active (e.g., during the daytime period), resulting in a reduction in seasonal cumulative ET between 1 and 51 mm, depending on the treatment. This is most likely a method specific extrapolation problem (see discussion) and the reason we use SVM for final budget estimations."*

L318-320 could be put at the top of the paragraph

*Done. It now reads: "In general, eroded soils tend to have a negative effect on $ET_{sum}$. However, this trend was not statistically significant (Kruskal-Wallis-Test, $ET_{sum}$: $\chi2 = 3.04$, df = 5, p = 0.69). DM and WUE, on the other hand, differed significantly between treatments (Kruskal-Wallis-Test, DM: $\chi^2 = 14.58$, df = 5, p = 0.01; WUE: $\chi^2 = 11.12$, df = 5, p = 0.05)."*

L321 this section could be merged with the previous – consider the treatment effect and then the drivers

*Done.*

L430 "to the amount of data" – how much? – you have a lot right? What would be sufficient?

*This statement refers to the general fact that, also according to cited literature, machine learning algorithms such as SVM and ANN perform better with larger datasets. While we*

*cannot quantify "sufficient data" for the failing ANN analysis, we can do so for the indeed working SVM through stepwise k-fold-subsampling.*

*It now reads: "Furthermore, it must be noted that the quality of SVM (and ANN_BR) predictions is highly dependent on the amount of data available (Chia et al. 2020; Abudu et al. 2010). Consequently, we tested the minimum amount of data necessary to provide predicted ET fluxes of 'good' quality (see criteria in M&M). For the particular dataset already 50 % of the total data available (minimum 300 measurements) provided 'good' results. Thus, we emphasize that capturing a large variability of fluxes under different environmental conditions seems to be more important than a merely large data set."*

L432 this section is very long- be more concise, move parts that cite other work to other parts of the discussion (or a new section there) – moreover, this section starts with a description of gap-filling – is that the main finding? If so the paper should be reworked so it's the dominant research question and the rest is a case study to test it. I'd tend to the think the focus should be on soils, management, and the resultant ET

*Thank you for this comment. We rewrote/restructured the conclusion. In regard to focus of the manuscript: we believe that both the methodological aspects (new measurement setup, gap filling) and the soil heterogeneity and dynamics in ET are equally important to address here. As mentioned above we have tried to clearly phrase this in the introduction and entirely reworked the discussion and conclusion sections to strictly stick to our defined aims and hypotheses.*

Fig 1 what do the colors in plot c represent? Where are the soil moisture measurements and adjacent field as indicated in text?

*The colors indicate the non-diluted and diluted plots of each soil type. We now included this information. Measurements of soil moisture were taken in the marked area (black circle, b). See also reply above.*

Fig 2 clarify that this is incoming par and not absorbed, reflected, etc.

*Done. It now reads: "Environmental parameters during the measurement period with (a) daily mean temperatures (orange line; light gray = corresponding variation) and daily mean relative humidity (black line; dotted lines = corresponding variation), (b) incoming photosynthetically active radiation (purple) and (c) soil moisture (blue line) and precipitation (blue bars)."*

Fig 3 change Okt to Oct, describe here or in the text where are values below zero? Indicate here or in the text whether these are already quality-screened and what those methods were, what is the estimated detection limit?

*Figure 3 shows only the measured ET fluxes quality-screened by soft and hard criteria without any gap-filled data. Due to the fact that negative values only occurred during gap filling with ANN-Br (see in detail above), there are no negative values in this figure.*

*It now reads: "The seasonal development (Fig. 3) of the quality-screened measured ET-fluxes is similar for all treatments: a short growth phase after germination (1 - 2 mmol m$^{-2}$ s$^{-1}$) is*

*followed by a decrease of fluxes until and during the non-growing season in winter (< 0.1 mmol m$^{-2}$ s$^{-1}$), when hardly any plant activity is found due to low temperatures."*

Fig 6 some stats perhaps could help tell us if these curves or cumulations are different, significantly

*We added the mean cumulative seasonal ET and the standard deviation to the figure.*

Fig 7 what are the colors?

*The colors describe the different treatments. We added a legend.*

Fig 8 is an nova possible here, which are specifically different from each other?

Since the data are not normally distributed, no ANOVA was performed. Results of statistical tests indicating a difference between treatments have been added.

*The paragraph in question now reads: "In general, eroded soils tend to have a negative effect on ET$_{sum}$. However, this trend was not statistically significant (Kruskal-Wallis-Test, ET$_{sum}$: χ2 = 3.04, df = 5, p = 0.69). DM and WUE, on the other hand, differed significantly between treatments (Kruskal-Wallis-Test, DM: χ$^2$ = 14.58, df = 5, p = 0.01; WUE: χ$^2$ = 11.12, df = 5, p = 0.05)."*

Fig A1 can real data be plotted here?
*We have added the average measured weekly data.*

**Technical comments**

L13 Add the before ongoing

*Done.*

L16 change ingoing to incoming (and throughout)

*Done.*

L17 the paper doesn't really address the full water budget – perhaps main water budget term would be better?

*That's right. We now call it "seasonal cumulative ET".*

L92, 114 change build to built

*Done.*

L95 change the aim was to the paper's aim is…

*Done.*

L102 reword "This enabled to asses"

*Done.*

L103-4 be consistent with WUE vs WUEagro

*Done.*

L114 the description of CarboZALF repeats the intro (and this sentence is too long)

*Removed this sentence. Done.*

L120 "organic fertilized" interrupts the flow; it can be omitted

*Done.*

L129 just topsoil (not topsoils) – and was not were

*We have rewritten this section for better understanding.*

L135 weighed not weighted

*We have rewritten this section for better understanding.*

L146 and elsewhere the "by" is not needed

*Done.*

L155 remove comma after both

*Done.*

L163 channel not canal

*Done.*

L166 write out three, clarify "10 seconds = 10 records"

*We have rewritten this section for better understanding.*

*It now reads: "Each plot was measured once a week for 30 seconds, resulting in one mean value including 30 measurement points."*

L174 perhaps replace "further called" "what we term the"

*We changed the formulation. It now reads: "During the period from November 24, 2020 to March 22, 2021, which we refer to as the "non-growing season",  no plant growth was assumed due to average daily temperatures below 5°C (<3 consecutive days)."*

L208 change I to 1 (i.e., "eye" to "one")

*Done. See next comment for citation.*

L208-210 consider re-organizing like: "a simple stat approach: (1) MDV), two empirical… (2, 3), and two machine learning.(4,5) for consistent structure

*Done.*

*It now reads: "Gap-filling procedures included: 1.) a simple statistical approach: "Mean diurnal variation" (MDV); two empirical approaches: namely 2.) "non-linear regression" (NLR) and 3.) "Look-Up-Tables" (LUT) as well as two machine learning approaches: with 4.) "Support Vector machine" (SVM) and 5.) "artificial neural network with Bayesian regularization"(ANN_BR)."*

L258 add values after NSE

*We added a table with values for different performance statistics (NSE, NRMSE, MAE, etc.) and thereon after refer to those.*

*It now reads: "Then, all the coefficient of determination (R2), mean absolute error (MAE), normalized root mean square error (NRMSE) and Nash-Sutcliffe efficiency (NSE) were calculated and used to define performance classes (Table 1) to evaluate the accuracy of the approaches for the given setup (Moriasi et al. 2015)."*

L259 remove "the" and the s in parameters

*Done.*

*It now reads: "To determine parameter impact on ET, linear and non-linear models were used. Lastly, differences of ET$_{sum}$, DM and WUE$_{agro}$ between treatments were tested with the Kruskal-Wallis-test."*

L269 change photosynthetic to photosynthetically

*We have now defined and abbreviated PAR before.*

*It now reads: "PAR (Fig. 2b) largely reflected the seasonal variation of the day length with a maximum of 1860 µmol m$^{-2}$ s$^{-1}$ (half hourly measurements), and reduced values during longer storm events and high cloud cover."*

L274 remove "in the observed period" or otherwise reword

*Done. It now reads: "The largest precipitation events (> 10 mm d$^{-1}$) occurred on September 26, 2020 with 12 mm, on December 24, 2020 with 15 mm and on February 3, 2021 with 16 mm."*

L275-6 to where does this downward trend go (Describe/quantify in text so one doesn't have to look at the figure to determine it)

*Done.*

*It now reads: "A sharp declining trend in SM and no response to precipitation events is evident from April (about 25 %) to harvest in June (about 12 %)."*

L282 has a unique date style compared to the rest of the text

*Done.*

*It now reads: "The maximum RVI values were all reached at a similar time (May 15, 2021 to May 18, 2021)."*

L285 "nearly no differences" – perhaps negligible?

*We have rewritten this section.*

L287 "clearly"?

*Removed.*

*We have rewritten this section.*

*It now reads: "Higher RVI values were already reached in non-eroded and strongly eroded soils compared to extremely eroded soil during the initial growing season in fall of 2020 until the non-growing season."*

L327 remove s from variables

*We have rewritten this section.*

L330 remove used

*We have rewritten this section.*

L331 change offers to offered

*We have rewritten this section.*

L333 consider fewer rather than less

*We have rewritten this section.*

L335 remove comma

*We have rewritten this section.*

L337 reword – no need for "As well as" twice

*Done. It now reads: "In the following, we will discuss i) the effects of soil-type and top-soil dilution on $ET_{sum}$, yield (DM) and $WUE_{agro}$, along with ii) the spatio-temporal variability of ET fluxes over the cultivation period, and iii) the suitability of the gap-filling strategies used in this study as well as potential ways forward to improve our approaches."*

L337 add of after suitability

*Done. See comment above.*

L365 change was to were

*We have rewritten this section.*

L397 change were spanning over to spanned

*We have rewritten this section.*

L402 add t to constraints

*Done. It now reads: "This is particularly true in light of ongoing discussions surrounding method constraints of estimating ET across scales (Ding et al. 2021; Ghiat et al. 2021; Hamel et al. 2015)."*

L445 remove comma after demonstrate

*We have rewritten this section. It now reads: "Here, we present a possibility to obtain not only plausible $ET_{sum}$ but also diurnal cycles of ET by using the novel FluxCrane system in combination with SVM based gap filling."*

The authors present evapotranspiration (ET), water use efficiency (WUE) and ET gap-filling strategies along a toposequence of erosion for one growing season of winter wheat in NE Germany. In addition, a comparison between topsoil addition/removal is included in the study. ET is measured with a novel automatic chamber system comprising large transparent cylindrical chambers which are lowered on a collar from above using a crane. The advantage of this system is the relatively large measurement area and less impact of the automatic chamber as compared to other chamber systems. The study can be recommended for publication, however, there are aspects which should be considered for revision. (1) a complete revision from a native English speaker would improve the readability of the manuscript (disclaimer: saying that without being a native English speaker myself). (2) The method section should include more information on how ET was measured. (3) The discussion and conclusion section would need complete revision; the prior needs more focus on the study, the latter is too long and should be condensed to the most important aspects. (4) the measured ET has not been compared to other ET methodologies; this discussion point should be further strengthened. See details below:

*We greatly appreciate your constructive review of our manuscript. We took your comments into careful consideration and you will find detailed replies below each comment marked in italics for better readability.*

*In regard to your general suggestions:*

1. *A native speaking colleague carefully read, spell check the revised version and advise us on writing style. We acknowledge his work in the revised version.*

2. *We agree that this has been kept rather short. We have taken this into account and expanded the Material and Methods section accordingly (see also specific comments below).*

3. *We again agree. In the revised version, the discussion section has been completely reworked. We have reworked our aims and included hypotheses and have strictly reworked the discussion section into subsections adhering to our aims and hypotheses. We also included the required additional sections, e.g. on methodological comparability. The conclusions have been significantly shortened, focusing now on the major outcome of our study.*

4. *This is true. Within the field of research, we have seen an intensive discussion on the methodological constraints of different approaches to measure and model ET. We have briefly touched this subject already in the discussion (relating our results to previous findings of a nearby lysimeter study), but now included a more detailed account on the subject. This includes comparisons to other studies and approaches but also comparing our results with estimates of potential and actual ET of the region during the study period.*

**Abstract**

L33: please include month of harvest here

*Done. (See next Comment for citation)*

L34: the concluding sentence is not clear to the reader, what is meant by "small long-term differences"?

*Small differences over the entire cultivation season are responsible for most of the differences in cumulative seasonal ET. It now reads: "The key period contributing to 70 % of $ET_{sum}$ was from the beginning of stem elongation in April to harvest in June. However, differences in the $ET_{sum}$ between soil types and topsoil dilution resulted predominantly from small differences between the treatments throughout the cultivation, rather than only during this short period of time."*

**Introduction**

L39-40: Please include a reference showing that climate change and increase in human population are responsible for increased land use for agriculture.

*Done. It now reads: "Much of this land is already in use to ensure food security, mandated by a still growing human population paired with the ongoing climate crisis (Searchinger et al. 2018)."*

L44/57: what is meant by "fertility" and "fertile" here? Probably better to use the term "plant production", "yield" "productive" etc. because fertility and fertile refers to generative reproduction

*Done. It now reads: "In hummocky landscapes, erosion and associated topsoil dilution caused by, e.g. wind, water or tillage, affects the crop yields (Bakker et al. 2007; Biggelaar et al. 2003)."*

L47: is it 10-40% of rainfall?

*Yes it is. It now reads: "of precipitation".*

L48-50: not sure, if this is a trend. NE Germany has typically lower rainfall compared to the rest of Germany, should not be related to climate change

*That's right. This sentence was confusing due to the fact that Germany is part of the region with increasing precipitation in general. We changed it for a better understanding.*

*It now reads: "In Germany, annual precipitation is more than 800 mm in most regions of west and south Germany but only 400 - 500 mm y$^{-1}$ in the northeast (e.g. areas in Brandenburg and Mecklenburg-Western Pomerania; Schappert, 2018)."*

L82: should probably read "field scale" here, using eddy covariance on plot level is difficult.

*Done.*

L83: if this is the footprint of eddy-covariance: please keep in mind that the footprint (in field crops) is probably lower than millions of square meters – especially at daytime (which is the important time period for ET) and during the growing season. The measurement height must be quite high then to reach such a huge footprint.

*It now reads: "At the field scale for example, eddy covariance systems provide high frequency estimates of ET of a homogeneous system while currently dominant manual chamber approaches are able to precisely capture multi treatment effects (<1m²) on ET at the plot scale, but lack the high frequency."*

L103: include "WUEagro" in brackets here.

*Done.*

L109: should read "soil surface"

*Done. (Not part of the manuscript any more)*

L113: should read "study" instead of "investigations"

*Done.*

L124 and following: please remove brackets where not needed.

*Done.*

L126 and following: probably better to use the term "topsoil modification" as in L121.

*We now call it "top soil dilution".*

L129/140: should read "fertilizer application" instead of "fertilization"

*Done.*

L129: should probably be 3 of 6 plots per soil type

*Done.*

L130-135: it is not clear to the reader why this modification of the soil was done? What is the reason for it?

*In the present manuscript we highlight the feedbacks of topsoil changes on the evapotranspiration. We have expanded this aspect in the revised version accordingly, particularly in the reworked discussion.*

*It now reads: "The manipulative field experiment was originally established to study the feedbacks of a dynamic disequilibrium in the carbon cycle of arable lands. Deep tillage or soil erosion lead to an admixture of subsoil material into the plough layer (Doetterl et al. 2016) which alters topsoil properties (SOC, clay content etc.)."*

What can be taken from the comparison of modified vs. non-modified?

*From the comparison of modified vs. non-modified we obtain new knowledge on crop growth and evaporation feedbacks after a strong erosion event or a deep ploughing measure.*

*It now reads: "The resulting changes in the main rooting zone might reduce crop growth (Öttl et al. 2021). We mimic these common landscape processes in our top-soil dilution experiment under controlled conditions (Vaidya et al. 2021)."*

This needs further explanation in the Introduction, Method, and Discussion section.

*It now reads: "After topsoil removal (1.2 t per plot; first 8-9 cm; 3 of the 6 plots per soil; July 14-15, 2020) we added the equivalent mass (1.2 t) of the respective subsoil horizons (E, Bt, Ck) taken from a large soil pit nearby. Thus, E horizon was applied to the prepared plots of the non-eroded Calcic Luvisol (LV-cc), Bt horizon on the strongly eroded Nudiargic Luvisol (LV-ng) and Ck horizon to the extremely eroded Calcaric Regosol (RG-ca)."*

L134: How much soil was added to each of the soil types?

*See comment above!*

L140: what was the amount of macronutrients applied?

*We included a table with detailed fertilization information in the appendix and refer to it in this section.*

*It now reads: "Fertilization was applied to all plots per soil type before and during the growing season using digestate from Pflanzenbauhof GbR (Uckermark, Germany), Triple Super Phosphate (TSP) and grain potash (Table B1)."*

L141-142: could this be a problem when comparing non-modified vs. modified plots?

*Due to initial changes in the topsoil structure (after adding subsoil material) germination differed between manipulated and non-manipulated. To achieve similar plant densities in all plots we replanted seedlings as described. By doing so, we avoid pure germination effects on the processes / fluxes studied.*

*We explain this now in the revised manuscript: "In order to achieve similar plant densities in all plots, replanting had to be done in all non-diluted plots within the frames (LV-cc: 13 plants per plot; LV-ng: 40 plants per plot; RG-ca: 82 plants per plot). "*

Did the modification improve germination of the crop?

*The differences in seed emergence could not be attributed exclusively to differences in manipulation/dilution. See comment above for more details.*

L143: When was glyphosate applied? The term "Round-up" can be removed.

*Done. It now reads: "For general plant protection and soil treatment, herbicides were applied to the field prior to the growing season (e.g. glyphosate; September 3, 2020)."*

L144-152: It would be recommendable to explain the whole system here, i.e., analyzers and sensors, variables taken, measurement rate, length of measurement, how many measurement per day etc.

*We agree that some information is missing or mentioned in the wrong place. All information concerning the system (mainly from chapter "2.5.1 ET Flux calculation") and other missing information are incorporated now in this sub chapter.*

L144: Did the chambers have fans?

*Two small axial flow fans (5.61 m3 min-1) were used to homogenize the chamber headspace air. This information is now included in the description of the gantry crane.*

L150-151: not sure what is meant here with "parallel"

*Both chambers measure simultaneously/parallel with a fixed distance, so that always the same two plots of the two rows are measured. Since the "parallel" here is confusing/obsolete, we have removed it.*

*It now reads: "Since the two chambers do not move independent from each other along the track, frames where arranged in rows, from which each half was measured by one chamber."*

L155-159: please include which sensors were used. Also, a discussion on inside and outside PAR and Temp is helpful here.

*We added all information about the Temp/PAR sensors and the effects on inside/outside measurements now.*

*It now reads: "The chambers have an average light transmittance of about 76 % (74% for chamber 1 and 78% for chamber 2), but a reduction in ET due to reduced light availability is not expected (Pape et al. 2009). Temperature differences during chamber closure were minimized by the short measurement time and ventilation (<1.5°C) with two small axial flow fans (5.61 $m^3$ $min^{-1}$) used to homogenize the chamber headspace air."*

L159: Is the SM measurement comparable when measured in an adjacent field without the treatment effects?

*Sorry for the misleading description, it was not an adjacent field it was a plot on the same field. We have now marked the adjacent plot in Fig. 1b (< 25m away). We added the location to the site map.*

L161: should read "plots"

*Done.*

L166 the area of measurement probably depends on the measurement height here, please include.

*That's correct. We added the fixed height.*

*Now it reads "The upward sensor was fitted with a cosine-correction diffusor for measurements of the incident radiation, while the downward sensor, installed 1.8m above the ground, had a 25° cone field of view, thus covering an area of 0.5 $m^2$ during measurements (Görres et al. 2014)."*

L166/167: it is not clear how many measurements were taken with this information, please re-phrase.

*Done. In now reads: „Each plot was measured once a week for 30 seconds, resulting in one mean value including 30 measurement points. "*

L172: Please describe further what is meant by "stagnation".

*Stagnation occurred during the winter due to plant inactivity (no plant growth).*

*It now reads: "Since only weekly plot-wise RVI data were available, daily RVI data were obtained by fitting a sigmoidal function for initial plant growth in the fall up to stagnation due to plant inactivity in the winter and a polynomial function for shoot elongation and later senescence during spring growth and summer maturation, respectively (Fig. A1)."*

L182-183: Please include the exact number of measurements. It is not clear how many measurements per day were made, how often was measured, how much original data was available.

*Done. In now reads: "A complete data set of hourly data points for the 286 days of the cultivation period would consist of 6888 measurements per treatment. After the flux calculation and filtering using the soft and hard criteria, a total of 13,011 ET flux measurements were performed, resulting in approximately 2,169 measurements per treatment. For individual plots, an average of 723 (624 to 1210; 10.5 %) measurements were measured and the remaining were predicted by the gap approaches (Table B2; 89.5% on average)."*

L184: Were nighttime data included in the calculations? This may impact mean values.

*All fluxes (day and night) of sufficient data quality (see section on hard and soft criteria) are included.*

*It now reads: "Diurnal ET day- and nighttime fluxes considered in this study were calculated for the cultivation period from September 17, 2020 (sowing of winter rye), until harvest of winter rye on June 30, 2021."*

*It is true that this affects the diurnal/seasonal ET in comparison to approaches where only daytime data is available making this rather a benefit of the approach. We now explicitly mention this in the discussion as follows:*

*"Moreover, the ability to observe nighttime fluxes has great potential to study previously overlooked short-term dynamics in ET and to improve the representation of underlying processes in process-based hydrological modeling, compared to other measurement systems and especially to manually operated chambers."*

L189: please include the step how ET was converted from mmol to kg (for WUE).

*We have now added an equation for the conversion:*

$$ET_{flux}[mmol \ m^{-2}s^{-1}] = \frac{ET_{flux}\left[\frac{mm}{d}\right]}{(t \times 1000)} * \left(\frac{1}{M_{H2O}}\right)$$

L195-197: please include how much data was there originally, how much data was excluded as outlier per treatment, and later, how much data was gap-filled.

*We included a table (Table B2) in the appendix with information on the number of measurements per treatment and the corresponding gap-filling percentages.*

L197 please write out "interquartile range"

*Done.*

L206: should read "2.5.2"

*Done.*

L244/311: why use SVM alone and not all gap-filling strategies and discuss it?

*We did all the calculations for the five different gap-filling strategies. We added plots like figure 6 for all methods to the supplement (Figure A3-6).*

L250: why not include NEE/GPP here and calculate WUE as GPP/ET? The LI850 would give the corresponding CO2 flux for it? The PAR and Temp could help to calculate respiration.

*We fully agree that a detailed account WUE based on detailed gas exchange not only of ET but also of NEE and GPP is absolutely relevant. Of course the LI850 is capable and indeed we assess CO2 cycling as well as its partitioning (see e.g. Vaidya et al., 2021) along with ET measurements. For the sake of a reasonable word limit and given the already dual focus (1. Gap filling of ET and 2. Impact of erosion and land-use on ET as much as it is visible from the first year of available data) of the present manuscript, we have decided to explore the treatment based differences as well as short term dynamics of WUE based on gas exchange of both carbon and water fluxes in a separate manuscript (once we have multi year gas exchange data for CO2 and water fluxes available). We profoundly belief that this is necessary to be able to holistically discuss the expected system responses in water use efficiency.*

L259-260: would a multilinear model work to estimate/explain ET? Is there difference among treatments or can the data be combined?

*A multilinear model would represent the NLR approach implemented already (including RH, Temp., RVI and soil moisture, PAR), which however performed less well than the SVM to estimate ET. Difference between relations to single parameters are shown within the figure in terms of different colored regression lines.*

**Results**

L264-265: should be part of Method section

*Done.*

L283-291: not sure about the approximation sign here ~: RVI values are shown with accuracy of two digits.

*That was a mistake. We were talking about the mean values of all three treatments.*

*It now reads: "In this regard, the non-diluted non-eroded soil "LV-cc n-d" had the highest RVI (16.46 on average), while the diluted non-eroded soil "LV-cc d" showed lower values (13.88 on average). The strongly eroded soil of "LV-ng" revealed the same pattern with a higher RVI for non-diluted (12 on average) and lower RVI on diluted (10.35 on average) treatments. The extremely eroded soil of "RG-ca", on the other hand, showed huge differences between the non-diluted and diluted treatments (10.95 vs. 5.87 on average)."*

L293-295: should be part of Method section, and in more detail.

*Done. See comment above for L182-183.*

L296-300: please include actual ET values here.

*We have added true values from ET in this section (3.3 ET Fluxes).*

*It now reads: "The seasonal development (Fig. 3) of the quality-screened measured ET-fluxes is similar for all treatments: a short growth phase after germination (1 - 2 mmol m$^{-2}$ s$^{-1}$) is followed by a decrease of fluxes until and during the non-growing season in winter (< 0.1 mmol m$^{-2}$ s$^{-1}$), when hardly any plant activity is found due to low temperatures. After the non-growing season, fluxes quickly return to their maximum fall level (1 - 2 mmol m$^{-2}$ s$^{-1}$) and then increase rapidly (> 5 mmol m$^{-2}$ s$^{-1}$). On the non-eroded soil (LV-cc), this rapid increase continued into June, while ET fluxes on the eroded soils (LV-ng and RZ-ca) already peaked in May. In addition, there is a clear difference in the maximum fluxes measured between soil types with 6.7 mmol m$^{-2}$ s$^{-1}$ for both treatments of non-eroded LV-cc, 5.6 / 6.5 mmol m$^{-2}$ s$^{-1}$ (n-d / d) for LV-ng, and 5.8 / 5.1 mmol m$^{-2}$ s$^{-1}$ (n-d / d) for RG-ca. Notably, there is a data gap from late April to late May due to sensor failure. "*

L308: what is meant by "allocation problems"?

*LUT allocation problems occur when no class is found for certain environmental parameters. The LUT approach will then use the mean value. We changed the formulation.*

*It now reads: "Considering the k-fold cross-validation (Fig. 5, Table 3), ANN_BR and SVM perform good, while MDV shows partially satisfactory statistics, and LUT shows unsatisfactory statistics due to allocation problems that arise when no class is found for the given climate conditions and the mean is used."*

L309: what is the difference between Table 1 and 2. The table description does not tell.

*Table 1 shows the calibration statistics and Table 2 shows the validation statistics as explained in the description. We have changed the description to calibration/validation to avoid confusion.*

L311: could it be a solution to set negative values to zero? Were these nighttime values?

*Computationally we could set the values to zero. However, this would violate general assumptions about the possibility of occurrence of negative ET fluxes. We now added a more detailed explanation:*

*"However, gap-filled ET fluxes using ANN_BR showed a large number of predicted negative ET fluxes (1547 on average per plot; Figure A6) throughout the cultivation period. These fluxes occurred to an unrealistic degree during times when RH was significantly below saturation and plants were active (e.g., during the daytime period), resulting in a reduction in seasonal cumulative ET between 1 and 51 mm, depending on the treatment. This is most*

*likely a method specific extrapolation problem (see discussion) and the reason we use SVM for final budget estimations."*

L319-320: should be part of section 3.6.

*We have now merged the two sections as suggested by reviewer one.*

**Discussion**

L328: Micro Bowen ratio systems have a relatively small footprint and can be used for small scales.

*It is true that the Micro Bowen Ratio is not appropriate as an example at this point. We deleted it.*

L330: "used, new" please re-phrase: the reader may think: "either it is used or new"

*We have rewritten this section.*

L332: Please re-phrase, because eddy covariance ET flux is averaged over 30-minute intervals and measures at 20 Hz.

*This paragraph has been shortened, making the sentence no longer part of the manuscript.*

L334: it is not clear from the current version of the manuscript, if the ET is measured reliably (I suspect so, but it is not clear to the reader at this point). Please include a Discussion section about it.

*We agree. While we have tried to discuss this to some degree (see the relation to a nearby lysimeter study, which however can´t be directly compared since the data is from previous years with different crops), this section was not very extensive and did not give the subject enough credit. We have now included a much more detailed discussion of this subject, trying to compare our results with previous studies, as well as with potential and actual ET for the region/period studied (see general comment 4).*

L340: it is not clear what "long term" means here.

*"Long-term" here means "over the entire cultivation season". We have changed it for better understanding.*

*It now reads: "Over the entire cultivation period, ET fluxes responded particularly to crop growth, first during the establishment period in fall (mid of October to mid of November) and then again during the main growth period in spring (end of March to mid of May)."*

L244: it is not clear what "more dynamically" means.

*In this context, "more dynamically" describes the naturally occurring diurnal effects on ET due to temperature and relative humidity, compared to the effects of RVI (plant growth), which occur slowly over the entire cultivation season. We have now explained this in more detail now.*

*It now reads: "Over the diurnal cycle, ET reacted to changes in environmental conditions, particularly temperature and RH, which together define the vapor-pressure deficit (VPD), as well as PAR."*

L345: why not include VPD in the study here. It should be possible to calculate it. It may give more insight the crop-atmosphere interaction.

*It is indeed possible to calculate the gap filling approaches using the vapor pressure deficit (VPD), but temperature and relative humidity should be excluded to avoid autocorrelation. We calculated all the gap filling approaches using VPD, but upon comparison, we found that the calculation using temperature and relative humidity gave more accurate ET estimates.*

L346: how does PAR impact ET? There is significant relationship, but an explanation is missing.

*PAR has two effects on ET: increasing PAR leads to both an increase in photosynthesis and affects stomatal conductance, and an increase in temperature on the surface of the plants. This leads to an increase in ET with increasing PAR. It is now included in the manuscript.*

*It now reads: "PAR (Fig. 2b) largely reflected the seasonal variation of the day length with a maximum of 1860 µmol $m^{-2} s^{-1}$ (half hourly measurements), and reduced values during longer storm events and high cloud cover (e.g. through changes in photosynthesis)."*

L347: exclude "in summary"

*Done.*

L349: only one measurement from an adjacent field available.

*That is correct, we have now discussed the implications of our limited soil moisture data set in more detail.*

*It now reads: "In this regard, using plot specific multi-depth SM data could also improve the predicted $ET_{sum}$ based on SVM in the future."*

L382-383: what is meant by "static differences"?

*In this context, "static differences" means differences that develop slowly over the cultivation season such as plant growth.*

L395: should read "ET"

*Done.*

L395-398: should be part of a separate section on the accuracy of the ET measurements.

*Yes, we agree. We have now added a separate discussion section on accuracy (4.4 Accuracy of the new system) that covers this statement.*

L405-410: please re-phrase this statement.

*We have rewritten this section.*

L427-429: why not include wind speed in this study? Were no measurements available? Strong winds are mentioned L447, so maybe the crane is equipped with an anemometer?

*The crane is equipped with an anemometer and wind speed data are available. However, we could not find any reasonable effect on ET during our study period. We added this check to our routine and will continue to check correlation between wind speed and ET.*

L412: please adjust LUT then for this study. There is room for improvement, as can be seen in Figure 5. It seems that a few events are off, which if excluded improve the R2 substantially.

*This sentence was misleading. Studies with other datasets (as cited here) could produce valid results with the LUT. With our dataset, the only way to avoid allocation problems was to use fewer classes. This resulted in a loss of variability, making diurnal differences invisible and seasonal ET estimates less accurate. Points in figure 5 show the allocation problems and are thus a vital result of the validation.*

*It now reads: "While the LUT showed very good calibration results, the allocation problems that occur when no class is found (Fig. 5) and the mean is used resulted in the lowest predictive ability during validation over the full range of measured ET fluxes. Some studies also obtained quite plausible results for LUT and MDV (Boudhina et al. 2018; Falge et al. 2001a; Moffat et al. 2007), and adjusting the classes of the LUT could further improve the results of this approach. However, with the available dataset, the only way to avoid allocation problems was to use fewer classes. This resulted in a loss of variability, making diurnal differences invisible and ET estimates less accurate."*

Conclusion – in general too long and rather a discussion than a conclusion. Please consider moving L433-444 and L454-464 (and others) to the discussion section, and re-write the conclusion as a summary of the most important findings.

*This section has been rewritten accordingly.*

L466: Why not include it in this study?

*It is not entirely clear to us which statement this refers to. We did not have any further data available in regard to either long-term ET data or the Measurements of stable water isotopes.*

Figure 3: "replicates summarized" – is it the mean or the sum of three replicates?

*We removed "replicates summarized". Figure 3 shows all the ET values calculated from the measurements of all three replicates per treatment.*

Table B2 and Figure 7 could be combined.

*This is a great idea. We combined them now.*

Figure 8. what is the standard deviation and what is the mean in the figure? Also, please discuss the effect of DM and ET on WUE in the Discussion section in more detail. E.g., it seems quite interesting that modified and non-modified Regosols have similar ET but different DM which impacts WUE.

*We have now reworked this plot to better reflect the mean and standard deviation. The effect of DM and ET on WUE was discussed in more detail in the discussion section "4.2 Seasonal variability of ET fluxes and WUE".*

Fiugre A1/Figure 6: what kind of variation is shown in the graph?

*In both figures, the lines describe the mean values of the three replicates with the shadow (variation) as the standard deviation. We have changed the description now. It now reads:*

*"Figure A1: RVI fit (colored lines) of the different treatments with the standard deviation between replicates (light gray) and the corresponding averages of the daily measurements (points)."*

*"Figure 6: Daily mean ET sums (colored lines) of the different treatments and seasonal cumulative ET ($ET_{sum}$; dashed lines) with standard deviation between replicates (light and dark gray)."*

Figure 6: seasonal ET, is this the cumulative ET shown here?

*Yes it is the "seasonal cumulative ET". We use this term here and in the whole manuscript.*

---

## Author Response (AR2)

To Mariano Moreno de las Heras:

Attached please find our revised manuscript entitled "*Benefits of a robotic chamber system for determining evapotranspiration in an erosion affected, heterogeneous cropland*" to be considered for publication.

This work has been previously submitted to HESS, with "major/minor revisions" being suggested to apply to our manuscript. The main recommendations of both reviewers were to better discuss the proportion of measured vs. modeled data and possible uncertainties of our approach compared to other approaches estimating ET; improving our description of how calibration/validation were carried out.

We have substantially revised our manuscript according to these suggestions, taking into account all criticisms and comments. In terms of the amount of originally measured data it is true that the term "gap-filling" is misleading here, since we calculate our continuous half hourly ET fluxes (as well as the cumulative ET) as described with data predicted by our data-driven models which were calibrated and validated using our original measurements. For this reason, we have changed the terminology from "gap-filling" to "modeling" throughout the manuscript and additionally discuss the implications of our approach in more detail particularly in comparison to other approaches determining ET. Moreover, we now include a more detailed description of calibration and validation and a more detailed discussion on the advantages and disadvantages of our system.

In conclusion, we believe the implemented revisions improved the manuscript significantly and thank you very much for your consideration.

Sincerely,

Adrian Dahlmann

**Review of "Benefits of a robotic chamber system for determining evapotranspiration in an erosion affected, heterogeneous cropland" by Adrian Dahlmann et al., for HESS.**

**Summary**

This manuscript reads much more cleanly than the previous, with a clearer vision, better writing, and more concise presentation. The hypotheses are now explicit and the flow is greatly improved. While I see a number of places where the language can improve – both in grammar and also to be more concise – the work is stronger.

**Minor comments**

Consider making the intro more concise. I think the paragraph from L36-56 could easily lose 100 words and be more focused (e.g. focus on a few of the challenges in the first 5 lines, reduce reference to six degradation processes, avoid phrases like "indispensable prerequisite" , consider whether both "crop yields" and "related crop productivity" are needed alongside "agriculturally used".

*Thank you for this suggestion. We have now shortened the introductory section by 68 words.*

L60 I think T is not 90% of ET? Perhaps you mean ET is 90% of the water budget? Anyway it's known that E and T are important, I'm not sure how much detail we need here.

*It is true that this sentence is not mandatory. This sentence was removed as part of the shortening of the introduction.*

L140 consider starting a new paragraph at "in the following" and decide whether n-d and d are actually good labels (to me they are too short; it was difficult to go back and find this). (see also the complexity of L309; perhaps also more intuitive names could be given to the soil types)

*It is true that the n-d and d abbreviations are very short and are used continuously only at a much later stage. For better understanding and comprehension, we now use these abbreviations earlier and more frequently (starting in L148).*

*Since the abbreviations of the soil types have already been used in other publications, we would like to keep these names for better comparability.*

L161 why is a reduction in ET due to lower light not expected? In many landscapes there is a near 1:1 relationship between radiation and ET.

*It is true that there is a close relationship between the PAR and ET. Our chamber system is designed to minimize the impact on the physiological activity of the enclosed vegetation. To achieve this, the chambers are made of highly transparent materials with a low PAR reduction. In addition, in our experimental setup, the selection of the calculated flux closest to the start of the measurement, the short closure times, and the ventilation of the chamber should minimize physiological changes (e.g., stomatal responses) and thus minimize changes in ET fluxes.*

*In similar experiments with dark chambers and chambers of this material (unpublished data), we have found that gas fluxes do not differ between dark and light, and thus no plant response is detectable in the first 5 minutes (at least for the crops we considered). For this reason, we expected a negligible effect on calculated ET due to PAR reduction. We have now added this information to the manuscript.*

L165 I'm not sure "Sensor death time" is a clear concept

*We now call it "sensor response time".*

L275 or elsewhere can the Kc values be given?
*We now added the $K_c$ values in brackets.*

*It now reads: "$ET_c$ (Equ.6) was calculated from the crop factor Kc ($K_{c\ ini}$ = 0.3; $K_{c\ mid}$ = 1.15; $K_{c\ end}$ = 0.4) and the potential evapotranspiration ET0 using monthly averages (DWD 2022)."*

L330 and throughout this section some numbers would be helpful – how much overestimation and underestimation (Either in mm or relative error, for example).

*We have now calculated the mean error for the calibration in three different ET flux ranges (Table B4; less than 2, between 2 and 4, and greater than 4 mmol m-2 s-1) for all modeling approaches and refer to them in this section.*

L334 "perform good" is not grammatically correct and should be "perform well'; perhaps "scores in the good category" would help make it both align with Table 3 and be grammatically correct.

*Thank you. We now use "scores in the good category".*

Some parts of the discussion could be split into clearer paragraphs and more concise language could be used throughout.

*We have partially rewritten the discussion and added additional paragraphs for better understanding.*

L492 "diurnal cycles" are not really explored in this work; consider shifting the focus of this a bit (or even adding some more rigorous examination of diurnal cycling in the paper / appendix).

*We now present and discuss a new figure (Figure 6) with diurnal cycles in ET fluxes during the cultivation season for one sample day per month (day with the most measurements) and corresponding mean error (ME; two digits rounded).*

L500-505 isotopes is a strange ending; even the CO2 fluxes mentioned by the other reviewer could be a better ending since that data exists.

*We have now rewritten the conclusion to better represent what is already possible with the new system and what could be possible in the future.*

*It now reads: „In combination with $CO_2$ measurements, the novel FluxCrane could give new insights in ecosystem WUE in a high spatial resolution using NEE (net ecosystem exchange). In addition, coupled with the GEP (gross ecosystem production) and innovative measurements such as in-situ stable water isotopes (Dubbert et al. 2014; Kübert et al. 2020), a separation of ET into T and E would be possible to assess crop performance due to the plant specific WUE (Tallec et al., 2013) or study root water-uptake dynamics (Deseano Diaz et al. 2023; Kühnhammer et al. 2020). This is particularly relevant for regions with strong spatial heterogeneity in soils and generally low precipitation like the Uckermark and of crucial importance for the terrestrial water balance as well as the prediction of future ecosystem feedbacks (Groh et al. 2020)."*

The figure captions can be improved – e.g., Fig 1 (And the plot itself) start in the upper right but then are referenced in the order a-b-d-c (and never e);
*We now rearranged the figure.*

Fig 2 could separate T and RH
*Done.*

Fig 4-5 – can remove the word "bottom" and I think the treatments are on the left and the approaches are on the top (I would call gap filling an approach; soil experiments a treatment)
*Thank you. We now changed the description.*

**Technical comments**
L18-19, consider: "To do so, a 9-month plot experiment with winter rye was conducted at an eroded cropland located in a hilly moraine landscape in Uckermark, Northeast Germany."
*Done.*

*L33 drop the before ETsum*
*Done.*

L62-65 is too long.
*We have now shortened the first sentence there.*

L68-9, remove currently dominant (and consider some citations in this area)
*Done. It now reads: "At the field scale for example, eddy covariance systems provide high frequency estimates of ET of a homogeneous system (e.g., Ding et al. 2021) while manual chamber approaches are able to precisely capture multi treatment effects (<1m²) on ET at the plot scale (e.g., Hamel et al. 2015)."*

L155 add "ly" to independent.
*Done.*

L157 "further increase the chambers bearing surface" is not clear.
*We now rephrased this sentence.*

*It now reads: "To ensure airtight sealing during chamber deployment, steel frames with a diameter of 1.59 m and a depth of 5 cm were installed into the soil and equipped with an approximately 10 cm wide foam ring to further increase the chambers bearing surface, while deployed."*

L212 consider a new paragraph at "This procedure resulted in…"
*Done.*

L222 consider adding concentration before measurements.
*Done.*

L232 a space is needed before the parenthesis.
*Done.*

L249 no comma is needed after case.

*Done.*

The authors performed a complete review of the manuscript, and answered the reviewer's question comprehensively. The manuscript has been substantially improved, however, there are still a few aspects which need further clarification. One major point is the amount of original data left after data screening. If understood correctly, about 89% of the data originates from a gap-filled strategy. This is quite a substantial number to make an argument on treatment or temporal effects. Further details see below:

**Abstract**

L13-16: please use the introductory sentences to describe the statement of the problem here, e.g., the devastating impact of erosion and soil dilution on ET in NE Germany and the need to improve gap-filling strategies

*Done. It now reads:*

*"In light of the ongoing global climate crisis and related increases in extreme hydrological events, it is crucial to assess ecosystem resilience and - in agricultural systems - to ensure sustainable management and food security. In hummocky landscapes, erosion and associated topsoil dilution caused by wind, water, or tillage affect crop yields by, for example, reducing soil water storage capacity or decreasing rootability."*

L35 please include a concluding sentence here.

*Done. It now reads:*

*"In conclusion, the study could highlight effect of soil heterogeneity on dry biomass, evapotranspiration, and water use efficiency in agricultural systems, emphasizing the importance of considering soil characteristics to optimize crop productivity and resource management."*

**Introduction**

please compare L59 with the statement in L16. How are these two numbers related?

*Thank you. We found different numbers for this statement, but with the final quote it should be 100% in both cases.*

**Material and Methods:**

L161-162: Is PAR also reduced? If so, do you expect that ET is influenced in this study (despite the reference), owing to the significant relationship in L350-352, the gap-filling approach (L237/L242) and the discussion in L406?

*It is true that there is a close relationship between PAR and ET. For more details on the correlation between our system and PAR, please see the response to the other reviewer's comment.*

*Regarding the significant relationship between PAR and ET mentioned in the manuscript, we are only talking about the training and correlation of the model. Since we do not expect a reduction in ET due to PAR, these strong correlations do not affect the ET fluxes. We have now added this information to the manuscript.*

L225-227: 10% of original data is quite low, at least as compared to EC. Is this sufficient to understand treatment effects, diurnal changes of ET and WUE? Please consider this in your discussion.

*10% original data sounds very low at first glance. However, compared to other studies using chamber systems (e.g. Dubbert 2014), we were able to generate 7-10 times more data for each plot. However, it is true that the term "gap-filling" is misleading here, since we calculate our ET fluxes as described with data predicted by our models. For this reason, we have changed the terminology from "gap-filling" to "modeling" throughout the manuscript. In addition, we discuss this and additionally diurnal cycle (with the new figure 6) created with our data-driven model in more detail.*

Section 2.5.2: please include information on how calibration and validation was performed (e.g., was there a training and a validation set? How was this decided? Etc.)

*We now added a paragraph in section "2.7 Statistical analysis" with a detailed description of our calibration and validation.*

**Results:**
L346/Fig. 8: Is there a possibility to run a post-hoc test after Kruskal-Wallis? This would improve the understanding on the treatment effects.

*Thank you for your comment. We did the Dunn-Bonferroni test and included it in this section.*

*It now reads: "The subsequent Dunn-Bonferroni post-hoc test revealed only a significant difference in DM between non-eroded LV-cc n-d and eroded RG-ca d (p = 0.013). However, no statistically significant pairwise differences were found for WUE."*